



# Soil CO$_2$ efflux in an old-growth southern conifer forests (*Agathis australis*) – magnitude, components, and controls

**Luitgard Schwendenmann[1], Cate Macinnis-Ng[2]**

[1] School of Environment, University of Auckland,  Private Bag 92019, Auckland 1142, New Zealand

[2] School of Biological Sciences, University of Auckland, Private Bag 92019, Auckland 1142, New Zealand

Correspondence to: Luitgard Schwendenmann (l.schwendenmann@auckland.ac.nz)





## Abstract

Total soil $CO_2$ efflux and its component fluxes, autotrophic and heterotrophic respiration, were measured in a native forest in northern Aotearoa-New Zealand. The forest is dominated

by *Agathis australis* (kauri) and is on an acidic, clay rich soil. Soil $CO_2$ efflux, volumentric soil water content and soil temperature were measured bi-weekly to monthly at 42 locations over 18 months. Trenching and regression analysis was used to partition the total soil $CO_2$ efflux. The effect of tree structure was investigated by calculating an index of local contribution ($I_c$, based on tree size and distance to the measurement location) followed by

correlation analysis between $I_c$ and soil $CO_2$ efflux, root biomass, litterfall and soil characteristics. The mean total soil $CO_2$ efflux was 3.47 µmol m$^{-2}$ s$^{-1}$. Using uni- and bivariate models showed that soil temperature ($< 40\%$) and volumetric soil water content ($< 20\%$) were poor predictors of the temporal variation in total soil $CO_2$ efflux. In contrast, a stronger temperature sensitivity (around 57%) was found for heterotrophic respiration.

Autotrophic respiration accounted for 25 (trenching) or 28% (regression analysis) of total soil $CO_2$ efflux. We found significant positive relationships between kauri tree size distribution ($I_c$) and soil $CO_2$ efflux, root biomass and mineral soil CN ratio within 5-6 m of the measurement points. Using multiple regression analysis revealed that 97% of the spatial variability in soil $CO_2$ efflux in this kauri dominated stand was explained by root biomass and

soil temperature. Our findings highlight the need to consider tree species effects and spatial patterns in soil carbon related studies.

**Keywords:** autotrophic and heterotrophic respiration, collar insertion, organic layer, litterfall, root biomass, soil water content, soil temperature, tree structure, trenching, New Zealand





## 1 Introduction

Soil surface $CO_2$ efflux (soil respiration) is the largest $CO_2$ flux from terrestrial ecosystems
into the atmosphere (Raich and Potter, 1995; Janssens et al., 2001; Bond-Lamberty and
Thomson, 2010a). Quantifying the magnitude of soil $CO_2$ efflux and examining the spatial
and temporal heterogeneity of soil $CO_2$ efflux is critical in characterising the carbon (C)

dynamics in terrestrial ecosystems (Schlesinger and Andrews, 2000; Trumbore, 2006; Smith
and Fang, 2010) as even a small change in soil $CO_2$ efflux could have a strong impact on
atmospheric $CO_2$ concentration (Andrews et al., 1999; Rustad et al., 2000). Advancing the
understanding of soil $CO_2$ efflux and its driving factors is also important to predict the effects
of land-use conversion and climate change on the net C sink of the terrestrial biosphere

(Giardina et al., 2014).

Soil $CO_2$ efflux varies widely in space and time according to changes in various abiotic and
biotic factors. Across terrestrial ecosystems soil temperature is often the main abiotic factor
explaining temporal patterns of soil $CO_2$ efflux (Raich and Schlesinger, 1992; Jassal et al.,
2005; Bond-Lamberty and Thomson, 2010b). Many studies show a positive correlation

between soil temperature and soil $CO_2$ efflux and this relationship is often expressed as a $Q_{10}$
function (relative increase in soil $CO_2$ efflux rate per 10°C difference) (van't Hoff, 1898;
Lloyd and Taylor, 1994). However, other abiotic factors have been found to influence the
temporal and spatial variation in soil $CO_2$ efflux. For example, several studies have shown a
parabolic relationship between soil water content and soil $CO_2$ efflux with the highest soil

$CO_2$ efflux occurring at an intermediate soil water content (Davidson et al., 1998, 2000;
Schwendenmann et al., 2003). Other soil factors driving the variability in soil $CO_2$ efflux in
forest ecosystems include the quality and quantity of soil organic matter (Rayment and Jarvis,
2000; Epron et al., 2004) and microbial biomass (Xu and Qi, 2001).

Biotic factors that influence rates of soil $CO_2$ efflux include plant and microbial components.

Vegetation type and structure, are important determinants of soil $CO_2$ efflux because they
influence the quantity and quality of litter and root biomass supplied to the soil and they also
mediate the soil microclimate (Fang et al., 1998; Raich and Tufekcioglu, 2000; Metcalfe et
al., 2007). For example, litter addition experiments have shown that increasing litterfall
enhances soil $CO_2$ efflux (Sulzman et al., 2005; Sayer et al., 2011). A few studies have

investigated the effect of stand structure and tree size on soil $CO_2$ efflux in temperate
(Longdoz et al., 2000; Søe and Buchmann, 2005; Ngao et al., 2012) and tropical forests



(Ohashi et al., 2008; Katayama et al., 2009; Brechet et al., 2011). Findings demonstrate that the spatial distribution of emergent trees strongly affects the root distribution and litterfall, partly explaining the spatial variation of soil $CO_2$ efflux (Katayama et al., 2009; Brechet et

al., 2011). Some studies show that soil $CO_2$ efflux at the base of emergent trees is significantly higher compared to soil $CO_2$ efflux at greater distances from the trees (Katayama et al., 2009; Ohashi et al., 2008).

Soil $CO_2$ efflux is the result of $CO_2$ production by heterotrophic and autotrophic respiration and gas transport (Fang and Moncrieff, 1999; Maier et al., 2011; Maier and Schack-Kirchner

2014). Heterotrophic respiration mainly originates from microbes decomposing plant detritus and soil organic matter while autotrophic (= root/rhizosphere) respiration comes from plant roots, mycorrhizal fungi and the rhizosphere (Hanson et al., 2000; Bond-Lamberty et al., 2011). The relative contribution of autotrophic respiration to total soil $CO_2$ efflux varies widely (10-90%) depending on the type of ecosystem studied (Hanson et al., 2000; Subke et

al., 2006; Bond-Lamberty et al., 2011). Various methods (i.e. trenching, regression analysis, isotopic methods) have been developed to separate heterotrophic and autotrophic respiration under both laboratory and field conditions and are described in the review papers by Hanson et al. (2000), Kuzyakov (2006) and Bond-Lamberty et al. (2011). Separating total soil $CO_2$ efflux into autotrophic and heterotrophic sources is important to more accurately predict C

fluxes under changing environmental conditions as heterotrophic and autotrophic respiration respond differently to abiotic and biotic factors (Boone et al., 1998; Davidson et al., 2006; Brüggemann et al., 2011). For example, heterotrophic respiration was found to be more susceptible to seasonal drought in a *Pinus contorta* forest (Scott-Denton et al., 2006). Other studies showed that autotrophic respiration in more temperature-sensitive compared to

heterotrophic respiration and total soil $CO_2$ efflux (Boone et al., 1998; Högberg, 2010).

Soil $CO_2$ efflux has been measured in a wide range of mature and old-growth forests across the globe (Schwendenmann et al., 2003; Epron et al. 2004; Sulzman et al., 2005; Adachi et al., 2006; Bahn et al., 2010; Bond-Lamberty and Thompson, 2014). An exception to this are the southern conifer forests (but see Urrutia-Jalabert, 2015) including kauri (*Agathis australis*

D. Don Lindl. ex Loudon, Araucariaceae) forests in Aotearoa-New Zealand. Old-growth kauri forests are considered to be one of the most C-dense forests worldwide (Keith et al., 2009) with up to 670 Mg C ha$^{-1}$ in living woody biomass (Silvester and Orchard, 1999). Kauri is endemic to northern New Zealand (north of latitude 38°S) (Ecroyd, 1982) and is the



largest and longest lived tree species in the country. Kauri has significant effects on the soil

environment (Whitlock, 1985; Verkaik et al., 2007) and plant community composition (Wyse et al., 2014). Phenolic compounds in kauri leaf litter (Verkaik et al., 2006) and low pH values (around 4) (Silvester, 2000; Wyse and Burns, 2013) partly explain the slow decomposition rates of kauri litter (Enright and Ogden, 1987) which result in thick organic layers in undisturbed kauri stands (Silvester and Orchard, 1999).

Organic layers (= forest floor composed of leaves, twigs and bark in various stages of decomposition above the soil surface) are important C reservoirs (Gaudinski et al., 2000) and can be a considerable source of $CO_2$ efflux. Organic layers can also contain a large amount of roots which may result in increased soil $CO_2$ efflux (Cavagnaro et al., 2012). Mature kauri trees have an extensive network of fine roots which extends from the lateral roots into the

interface between organic layer and the mineral soil (Bergin and Steward, 2004; Steward and Beveridge, 2010). A recent study also showed that roots and root nodules of kauri harbour arbuscular mycorrhizal fungi (Padamsee et al., in press). Roots cololonized by mycorrhizal fungi have been found to release more $CO_2$ than non-mycorrhizal roots (Valentine and Kleinert, 2007; Nottingham et al., 2010).

However, it remains unknown how much soil $CO_2$ is released from these C-rich southern conifer forests and which factors are driving the temporal and spatial variability in soil $CO_2$ efflux. It has been shown that kauri has a significant influence on soil properties but the influence of kauri tree distribution on soil carbon related ecosystem processes remains untested. Quantifying the magnitude of soil C loss and identifying the controls of this

significant C flux are essential for the assessment of the C balance of these C-rich and long-lived forest stands.

The aim of this study was to determine the magnitude, components and the driving factors of soil $CO_2$ efflux in an old-growth southern conifer forest. The specific objectives of our study were: (i) to quantify total soil $CO_2$ efflux, (ii) to identify the factors controlling the temporal

variation of soil $CO_2$ efflux, (iii) to test the effect of kauri tree distribution on soil $CO_2$ efflux and soil properties, and (iv) to determine the contribution of autotrophic respiration to total soil $CO_2$ efflux. In order to achieve the objectives we measured soil $CO_2$ efflux in an old-growth kauri stand over 18 months. To separate heterotrophic and autotrophic respiration we used direct (trenching) and indirect (regression technique) approaches.




## 2   Material and methods

### 2.1   Study site

The study was conducted in the University of Auckland Huapai reserve. The reserve is a 15
ha remnant of forest surrounded by farmland (Thomas and Ogden, 1983) and is located

approximately 25 km west of central Auckland on the northern fringe of the Waitakere
Ranges (36° 47.7' S, 174° 29.5' E). Within the long-term research plot (50 x 40 m), the
diameter at breast height (DBH) of all trees ≥ 2.5 cm was measured, the species were
identified and their location mapped (Wunder et al., 2010) (Fig. 1). The plot is dominated by
kauri (770 stems ha$^{-1}$) with a basal area of 75 m$^2$ ha$^{-1}$, equating to approximately 80% of the

stand basal area (Wunder et al., 2010). Silver ferns (*Cyathea dealbata*) are also highly
abundant (785 stems ha$^{-1}$) (Wunder et al., 2010). Less-numerous species are a mixture of
podocarps and broadleaved species, including *Phyllocladus trichomanoides*, *Myrsine
australis*, *Coprosma arborea* and *Geniostoma ligustrifolium*.

Total annual rainfall, measured from 2011 to 2013 at a weather station located in the vicinity

of the reserve, is approximately 1200 mm with 70% occurring during austral winter (June-
August). Annual mean temperature is 14°C (Macinnis-Ng and Schwendenmann, 2015). The
soils are derived from andesitic tuffs and are classified as Orthic Granular Soils (Hewitt
1992). The clayey soil is fairly sticky when wet, and hard and fragile when dry (Thomas and
Ogden, 1983). The thickness of the organic layer varies between 5 and 15 cm and consists

mainly of partly decomposed kauri leaves and twigs.

### 2.2   Experimental setup

The long-term research plot was subdivided into six equal quadrats. Within each quadrant
two soil $CO_2$ efflux locations (in total 12) were randomly located (Fig. 1). For each location

we measured the distance to the closest tree with a DBH ≥ 2.5 cm. At each of these 12
locations, a cluster of measurements was made. There was one surface measurement and
three inserted measurements as described below.

Soil $CO_2$ efflux was measured on the surface of the forest floor by gently pressing a polyvinyl
chloride (PVC) ring attached to the soil respiration chamber (see below for details) down on

the forest floor during measurements to avoid cutting fine roots. The locations were marked
with flags and kept free of vegetation. Surface (= total) soil $CO_2$ efflux was measured over 18



months from August 2012 to January 2014 at each location. These sample points were named Plot_Surface.

Next to the locations for surface soil $CO_2$ efflux measurements, a cluster of PVC collars (10

cm in diameter, 20 cm in height) was inserted in November 2011 and left in place over the measurement period. Here after, these sample points are known as Plot_Inserted. Three collars per cluster were spaced evenly around the circumference of a circle 2 m in diameter, with small adjustments in the spacing to accommodate large roots. Each collar was driven right through the organic layer and 1-2 cm into the mineral soil layer to cut off the roots

growing in the organic layer. In order to prevent $CO_2$ uptake, any vegetation inside the collars was regularly removed. The thickness of the organic layer at each grid point was measured using a ruler outside each collar. Efflux was measured from January 2012 to January 2014.

We used the trenching approach to separate heterotroph and autotrophic respiration. To avoid

disturbing the long-term research plot the trenching experiment was set-up directly adjacent to the research plot. In July 2012, six 2 x 2 m plots were trenched to 30 cm depth based on a preliminary study showing that the majority of fine roots (over 80%) are located in the organic layer and top 30 cm of the mineral soil. The trenches were double-lined with a water permeable polypropylene fabric and backfilled. During trenching, trampling and disturbance

inside the 2 x 2 m plots were avoided as far as possible.

Three types of measurements were conducted in the trenched plots. First, surface soil $CO_2$ efflux was measured at one location outside each trenched plot (Outside_Trench_Surface) in the same way as the Plot_Surface samples were measured (see above). Second, a collar was randomly placed outside each trenched plot (Outside_Trench_Inserted) and third, two collars

were randomly placed inside the trenched plot (Trench_Inserted). The collars were inserted 1-2 cm into the mineral soil layer as described above. Soil $CO_2$ efflux was measured 1 day before and 1, 3, 5, 7, and 14 days after trenching and then bi-weekly to monthly until December 2013.

**2.3   Soil $CO_2$ efflux measurements**

Soil $CO_2$ efflux was measured with a portable infrared gas analyser (EGM-4, PP Systems, Amesbury, MA, USA) equipped with a soil respiration chamber (SRC-1, PP Systems,



Amesbury, MA, USA). The $CO_2$ concentration was measured every 5 sec over 90-120 sec between 9 am and 2 pm local time and the change in $CO_2$ concentration over time was

recorded. Diurnal soil $CO_2$ efflux measurements conducted in January 2013 showed that soil $CO_2$ efflux rates between 9 am and 2 pm were comparable as there was not siginificant diurnal trend (data not shown).

Soil $CO_2$ efflux ($\mu$mol m$^{-2}$ s$^{-1}$) was calculated as follows:

$$\text{Soil } CO_2 \text{ efflux } (\mu\text{mol m}^{-2}\text{ s}^{-1}) = (\Delta CO_2/\Delta t) \times (P \times V)/(R \times T \times A) \qquad (1)$$

Where $\Delta CO_2/\Delta t$ is the change in $CO_2$ concentration over time (t), calculated as the slope of the linear regression ($\mu$mol mol$^{-1}$ s$^{-1}$ = ppm s$^{-1}$), P is the atmospheric pressure (Pa), V is the volume of the chamber including collar (m$^3$), R is the universal gas constant, 8.314 m$^3$ Pa K$^{-1}$ mol$^{-1}$), T is the temperature (K) and A is the surface area of ground covered by each chamber (0.007854 m$^2$).

Soil temperature (Soil temperature probe, 10 cm probe, Novel Ways Ltd, Hamilton, New Zealand) and volumentric soil water content (Hydrosense II, 12 cm probe, Campbell Scientific Inc., Logan, UT, USA) were measured concurrently in close proximity to each of the collars.

**2.4   Litterfall, root and soil characteristics**

Litterfall (including leaves, twigs, fruits, flowers, cone scales, etc.) was collected from twelve litter traps (pop-up planters, 63 cm in diameter) located next to each soil $CO_2$ efflux cluster within the long-term research plot (Fig. 1). Litterfall was collected bi-weekly from January 2012 - January 2014, dried at 80°C until constant mass was achieved, sorted and weighed

(Macinnis-Ng and Schwendenmann, 2015).

Organic layer and mineral soil samples (0-10 cm depths) were taken next to each collar with a core sampler in November 2011 (research plot) and July 2012 (trenched locations). Samples were ground and analysed for total C and N concentration using an elemental analyser (TruSpec, LECO Corporation, St. Joseph, Michigan, USA). Soil (LECO Lot 1016, 1007) and

leaf (NIST SRM 1515 - Apple Leaves) standards were used for calibration. The coefficient of variation was of 0.5% for C and 1% N for plant material (45% C, 25 2.3% N) and 1% for C and N for soil (2 – 12% C, 0.2 – 1% N). 10% of samples were replicated and results were within the range of variation given for the standards.





Organic layer and mineral soil samples (0-15 cm, 15-30 cm) were collected for soil analysis
and root biomass estimation adjacent to clusters 1, 3, 5, 7, 10 and 12 and the trenched plots.
Organic layer samples were collected from 20 cm x 20 cm quadrats. Mineral soil samples
were taken using a 15-cm diameter steel cylinder. Samples were dried at 60°C (forest floor)
and 40°C (mineral soil). Mineral soil samples were sieved at 2 mm. pH was measured in a
1:2.5 soil-water suspension (SensION 3 pH meter, HACH, Loveland, CO, USA). The organic
layer samples were wetted and fine roots were manually picked with tweezers. Roots were
separated from the clay rich mineral soil by flotation. Roots were dried at 60°C until constant
mass was achieved and weighed by size class (fine roots: < 2 mm, and small (coarse) roots:
2-20 mm). Litterfall, root and soil data are summerized in Table 1.

**2.5   Data analysis**

The individual collar fluxes per cluster (Plot_Inserted, n=3) and the two replicates per
trenched plot (Outside_Trench_Inserted and Trench_Inserted) were averaged before further
statistical analysis. Further, data from each for the 12 (plot) and 6 sampling points outside the
trenched plots were averaged to calculate a mean for inserted samples for a particular
sampling date. Normality of the data distribution was examined using a Kolmogorov–
Smirnov test.

Two methods (trenching and regression-analysis) were used for partitioning of total soil $CO_2$
efflux. In the trenching approach, the trenched plus inserted (Trench_Inserted) treatment
represents heterotrophic respiration. Measurements from the soil surface (Plot_Surface and
Outside_Trench_Surface) represent total soil $CO_2$ efflux. Autotrophic respiration was
calculated as the difference between total soil $CO_2$ efflux and the efflux measured from the
Trench_Inserted locations. For the regression-analysis approach the heterotrophic respiration
was derived analytically as the y-intercept of the linear regression between root biomass
(independent variable) and total soil surface $CO_2$ efflux (dependent variable) (Kucera and
Kirkham, 1971; Kuzyakov, 2006). Autotrophic respiration was then estimated by subtracting
the heterotrophic respiration from total soil $CO_2$ efflux.

Spatial characteristics of soil $CO_2$ efflux, soil temperature and volumetric soil water content
were expressed using descriptive statistics (minimum, maximum, mean and median values,
standard deviation, standard error, coefficient of variation). Differences in soil $CO_2$ efflux
among treatments (Plot_Surface vs Plot_Inserted; Outside_Trench_Surfave vs



Outside_Trench_Inserted and Trench_Inserted) and seasons were tested using a mixed model where treatment was considered as a fixed effect and sampling dates as a random effect.

To explore the abiotic environmental drivers of soil $CO_2$ efflux, univariate and bivariate empirical models were used to quantify the relationship between soil $CO_2$ efflux, soil
temperature and soil moisture. The models included linear (Gupta and Singh, 1981), quadratic (Kirschbaum, 1995), $Q_{10}$ (Davidson et al., 2006; Fang and Moncrieff, 1999), polynomial (Schlentner and Van Cleve, 1985) and a modified Arrhenius function (Lloyd and Taylor, 1994) (Table 3). Data from within the research plot and data outside the research plot (in the trenching experiment) were analysed separately due to differences in the number of
locations and measurement frequency. Coefficient of determination ($R^2$) and root mean square error (RMSE) were used to evaluate model performance.

The influence of kauri tree size and distribution on surface soil $CO_2$ efflux, litterfall, root biomass and soil properties was tested using an index of local contribution ($I_c$). The $I_c$ index was calculated for each tree as a function of (1) the trunk cross section area (S) and (2) the
distance (d) from the measurement locations following the approach described in Bréchet et al. (2011). The following functions were tested: uniform, $I_c = S$); linear ($I_c = S \times (1-d/r)$); parabolic ($I_c = S \times (1-(d/r)^2)$)); exponential ($I_c = S \times e^{(d/r-d)}$) and power ($I_c = S \times (1-(d/r)^a)$)) where a is a coefficient of form and r is a fitted radius of influence (r, in m) (Brechet et al., 2011). It was assumed that all kauri trees had the same radius of influence (r,
i.e. the distance above which their contribution would become negligable). The relationships between litterfall, root biomass or soil $CO_2$ efflux and the sum of the $I_c$ were assessed by using the coefficient of determination as a criterion to select the best model.

The spatial variability in soil $CO_2$ efflux was quantified at the plot scale using the coefficient of variation. Multiple regression analysis was used to assess the spatial controls (soil
temperature, soil moisture, organic layer thickness, soil C and nitrogen, root biomass) of surface soil $CO_2$ efflux.

Descriptive statistics, mixed model and multiple regression analysis were performed using SPSS v. 22 (IBM SPSS Statistics, IBM Corporation, Chicago, IL, USA). The univariate and bivariate soil temperature and moisture functions were done using Matlab (Version
7.12.0.635, The MathWorks, Natick, MA, USA). The local contribution analysis ($I_c$) was conducted using R (v3.1.0 R Development Core Team, 2005). Significance for all statistical analyses was accepted at $p < 0.05$.



### 3 Results

#### 3.1 Treatment effects and seasonal variations in soil $CO_2$ efflux, soil
temperature and volumetric soil water content

During the study period, soil temperature and moisture varied with season (Fig. 2).
Summertime soil temperatures peaked at about 17°C while minimum winter temperatures
were around 11°C (Fig. 2) annual mean soil temperature was $14.2 \pm 0.1$°C (Table 1).
Volumetric soil water content (SWC) was highest during late winter/early spring with values
of 55% and soil was driest during late summer/early autumn with around 25% (Fig. 2).
Annual average was $43.9 \pm 0.9$% (Table 1). Across the study period, an average of $1.9 \pm 0.1$
kg m$^{-2}$ litter fell at the sampling locations and the organic layer was $8.8 \pm 0.9$ cm thick (Table
1). Other description information is summarised in Table 1.

Surface soil $CO_2$ efflux rates (Plot_Surface) measured at 12 locations within the research plot
varied from $0.7 – 9.9$ µmol $CO_2$ m$^{-2}$ s$^{-1}$ during the 18-month study period (Fig. 2). Surface
soil $CO_2$ efflux was positively skewed with the mean larger than the median (Table 2). The
mean surface soil $CO_2$ efflux ($\pm$ SE), averaged over the 12 locations and all sampling
locations, was $3.6 \pm 0.1$ µmol $CO_2$ m$^{-2}$ s$^{-1}$. Higher efflux rates were measured during austral
summer and early autumn (December-March, $2.7 - 4.7$ µmol $CO_2$ m$^{-2}$ s$^{-1}$) compared to winter
(June-August, $1.8 - 3.9$ µmol $CO_2$ m$^{-2}$ s$^{-1}$). However, differences among seasons were not
significant ($p > 0.05$). In contrast, soil temperature differed significantly between summer
(16.5°C) and winter (11.8 °C). We also detected significant seasonal differences in SWC with
drier conditions during summer (mean SWC = 31%) compared to winter (mean SWC =
47%).

Collar insertion had a significant effect on soil $CO_2$ efflux (Plot_Inserted, Table 2). Soil $CO_2$
efflux from inserted collars ($3.0 \pm 0.1$ µmol $CO_2$ m$^{-2}$ s$^{-1}$) was 17% lower compared to surface
soil $CO_2$ efflux ($3.6 \pm 0.1$ µmol $CO_2$ m$^{-2}$ s$^{-1}$) (Table 2). The overall temporal pattern (Fig. 2)
of soil $CO_2$ efflux was similar between inserted and surface collars (Fig. 2). However, soil
$CO_2$ efflux from inserted collars varied considerably during the dry summer in 2013. High
soil $CO_2$ efflux from inserted collars in April 2013 coincided with heavy rain events after a
long dry period with high litter input (see Macinnis-Ng and Schwendenmann, 2015 for
details). Despite significant differences in SWC and litter fall between summer/early autumn
in 2012 and the same period in 2013, we did not find significant differences in inserted collar
soil $CO_2$ efflux ($p > 0.05$) (Fig. 2).



Surface soil $CO_2$ efflux measured outside the trenched plots ranged from 0.6 to 6.9 µmol $CO_2$

$m^{-2}$ $s^{-1}$ with a mean of 3.1 ± 0.1 µmol $CO_2$ $m^{-2}$ $s^{-1}$ (Outside_Trench_Surface, Table 2). The

temporal pattern of surface soil $CO_2$ efflux was comparable between plot and trench

locations. However, the magnitude in surface soil $CO_2$ efflux differed between plot and

trench locations with lower rates measured in trench locations (Table 2). In contrast, no

significant differences were found in soil temperature (14.4 vs 13.2 °C) and SWC (44.7 vs

44.2%) between plot and trench locations (Table 2).

Similar to the findings observed for the research plot, inserted collar soil $CO_2$ efflux rates

(Outside_Trench_Inserted; 2.6 ± 0.1 µmol $CO_2$ $m^{-2}$ $s^{-1}$) were significantly lower (17%)

compared to surface flux (3.1 ± 0.1 µmol $CO_2$ $m^{-2}$ $s^{-1}$, Table 2). SWC was significantly

affected by collar insertion (Table 2).

Soil $CO_2$ efflux from Trench_Inserted collars was significantly lower (25%) compared to

surface soil $CO_2$ efflux (Table 2). However, differences in soil $CO_2$ efflux between the

Trench_Inserted (11% lower) and Outside_Trench_Insered were not significant (Table 2).

Volumetric soil water content in the trenched plots was significantly higher (56.8%)

compared to the untrenched locations (44%). In contrast, soil temperature was not

significantly affected by trenching (Table 2).

### 3.2    Contribution of autotrophic respiration to total soil $CO_2$ efflux

Mean autotrophic respiration derived from the trenching approach was 0.8 ± 0.1 µmol $CO_2$

$m^{-2}$ $s^{-1}$. The contribution of autotrophic respiration to total soil $CO_2$ efflux (to 30 cm depth)

was 25%. Excluding the roots from the organic layer through deep collar insertion showed

that roots in the organic layer contribute around 17% to total soil $CO_2$ efflux. The proportion

of autotrophic respiration to total soil $CO_2$ efflux tended to be lower during summer

(December – March) compared to winter (July – September). However, differences were not

statistically significant due to high variability in autotrophic respiration, especially during

summer (data not shown).

Surface (= total) soil $CO_2$ efflux (plot + trench; n = 18, mean = 3.47 µmol $CO_2$ $m^{-2}$ $s^{-1}$; SE =

0.20 µmol $CO_2$ $m^{-2}$ $s^{-1}$) was positively correlated with total root biomass to 30 cm depth ($R^2$ =

0.394, p = 0.042, intercept = 2.49 µmol $CO_2$ $m^{-2}$ $s^{-1}$) (Fig. 3). Using the regression approach

produced a autotrophic respiration estimate of 0.98 µmol $CO_2$ $m^{-2}$ $s^{-1}$. The proportion of





autotrophic respiration to total soil $CO_2$ efflux derived from the root biomass regression approach was 28%.

### 3.3   Effect of  soil temperature and volumetric soil water content on the
temporal variability in soil $CO_2$ efflux

Univariate linear regressions between soil temperature or SWC and surface soil $CO_2$ efflux for the Plot_Surface and Plot_Inserted sample points failed to achieve high $R^2$ values (Table 3). Using a quadratic temperature function explained around 42% of the temporal variation in surface soil $CO_2$ efflux. Bivariate polynomial and hyperbolic functions resulted in higher $R^2$

values ($R^2 = 0.537$-$0.585$) compared to univariate models (Table 3). However, the root mean squared errors (RMSE) for polynomial and hyperbolic functions were high compared to the other models implying a poorer fit. A considerably stronger soil temperature-soil $CO_2$ efflux relationship was found for the inserted collars. Soil temperature explained up to 57% of the variance of soil $CO_2$ efflux emitted from inserted collars (Table 3).

Volumetric soil water content explained less than 18% of the temporal variability in soil surface $CO_2$ efflux (Table 3). The quadratic function showed that volumetric soil water content was positively related with soil $CO_2$ efflux only when it was below 40%. Above 40% the correlation between volumetric soil water content and soil $CO_2$ efflux was negative.

Univariate linear and non-linar regressions for the Outsite_Trench_Surface,

Outside_Trench_Inserted, and Trench_Inserted sample points resulted in very low $R^2$ values, especially for the surface flux and inserted collars. A weak response of soil $CO_2$ efflux to soil temperature ($R^2 = 0.233$ - $0.271$) was found in the trenched plots (Table 3). The small sample size (n = 6 locations) may explain the lack of strong correlations for these treatments.

**3.4   Spatial variation in surface soil $CO_2$ efflux and environmental factors**

The spatial variability of surface soil $CO_2$ efflux between the 12 locations in the research plot was relatively high, with a coefficient of variation (CV) of 43% (Table 1).

We found a good relationship between the tree local contribution index ($I_c$,) and soil $CO_2$ efflux (Fig. 4.1b). The relationship was strongest (coefficient of determination, $R^2 = 0.342$,  p

$= 0.030$, linear model) within a radius of 5 m (Fig. 4.1a,b).





The spatial variation in total root biomass (0 - 30 cm depth; 0.9 to 8 kg m$^{-2}$) was very high (CV > 95%, Table 1). Similar to soil $CO_2$ efflux, a radius of 5 m provided also the best correlation between root biomass and $I_c$ (Fig. 4.2b). The coefficient of determination was $R^2$ = 0.985 (p = 0.021, univariate model, Fig. 4.2a,b).

Compared to root biomass and soil $CO_2$ efflux the spatial variation in litterfall (total amount over the 18-month period, 1.1 – 2.2 kg m$^{-2}$, Table 1) was small (CV = 20%, Table 1). We did not find any significant correlations between litterfall and $I_c$ (data not shown).

Between 8 and 29 kg C m$^{-2}$ were stored in the 6 - 12 cm thick organic layer (Table 1). C:N ratio differed considerably between the organic layer (31-58) and mineral soil (13-19).

Differences in pH were greater among locations compared to differences between organic layer and mineral soil (Table 3). Except for C:N ratio in the mineral soil ($R^2$ = 0.655, p = 0.000, linear  model, Fig. 4.3a,b), no correlations were found between $I_c$ and soil characteristics.

Using multiple regression analysis revealed that most of the spatial variability in surface soil
$CO_2$ efflux within the plot could be explained by soil temperature and root biomass ($R^2$ = 0.977, Adjusted $R^2$ = 0.953, F = 41.972, p = 0.023).



## 4 Discussion

### 4.1 Soil surface $CO_2$ efflux: magnitude and temporal variation

Mean soil surface $CO_2$ efflux ($3.47 \pm 0.2$ µmol $CO_2$ m$^{-2}$ s$^{-1}$; $1315 \pm 77$ g C m$^{-2}$ yr$^{-1}$) measured

in this kauri dominated forest was higher than mean values from mature conifer and mixed
conifer-hardwood temperate rainforests along the Pacific coast of North America (500 - 2300
g C m$^{-2}$ yr$^{-1}$; mean: $1100 \pm 65$ g C m$^{-2}$ yr$^{-1}$; n = 55) (Campbell and Law, 2005; Hibbard et al.,
2005; Bond-Lamberty and Tompson, 2014) and southern conifer (*Fitzroya cupressoides*
forests in southern Chile (500 - 800 g C m$^{-2}$ yr$^{-1}$; Urratia-Jalabert, 2015). Soil $CO_2$ emissions

from the kauri stand were also higher than efflux rates measured in other New Zealand
forests. For example, approximately 1000 g C m$^{-2}$ yr$^{-1}$ g were measured in a rimu
(*Dacrydium cupressinum,* conifer) dominated podocarp forest in South Westland (Hunt et
al., 2008) and annual soil $CO_2$ efflux in *Leptospermum scoparium/Kunzea ericoides* var.
*ericoides* shrublands ranged between 980 and 1030 g C m$^{-2}$ yr$^{-1}$ (Hedley et al., 2013). In

contrast, our values are within the range of values reported for mature unmanaged tropical
moist broadleaf forests (900 -2000 g C m$^{-2}$ yr$^{-1}$; mean: $1336 \pm 70$ g C m$^{-2}$ yr$^{-1}$; n = 27) (Raich
and Schlesinger, 1992; Schwendenmann et al., 2003; Bond-Lamberty and Tompson, 2014).

Our finding suggests that soil $CO_2$ efflux in a conifer dominated forest can be as high or even
exceed the efflux rates from broadleaf forests. This is in contrast to previous studies which

found that soil $CO_2$ efflux in conifer forests are lower than those in broadleaf forests (Raich
and Tufekcioglu, 2000; Curiel Yuste et al., 2005). However, these studies were limited to
temperate locations and based on direct comparisons of sites where forest type was the
principal variable differing among pairs. Mean annual soil temperature has been shown to be
a good predictor of large-scale variation in total soil $CO_2$ efflux in non-water limited systems

independent of vegetation types and biome (Bahn et al., 2010). With a mean annual
temperature of 14°C this study site was relatively warm compared to sites along the Pacific
coast of North America partly explaining the high soil $CO_2$ efflux rates in this kauri
dominated forest.

The amount of litterfall has also been associated with differences in soil $CO_2$ efflux at the

scales of biomes (Davidson et al., 2002; Reichstein et al., 2003; Oishi et al., 2013). Annual C
input via litterfall in this kauri dominted forest was 410 and 760 g C m$^{-2}$ in 2012 and 2013,
respectively (Macinnis-Ng and Schwendenmann, 2015). This litter C flux is substantially
higher than those values from conifer and mixed conifer-hardwood forests in the Northern





Hemisphere (50 - 400 g C m$^{-2}$ yr$^{-1}$; mean: 164 ± 14 g C m$^{-2}$ yr$^{-1}$; n = 43; Bond-Lamberty and

Tompson, 2014; Holland et al., 2015). Kauri litterfall is within the range of values (110 - 700

g C m$^{-2}$ yr$^{-1}$; mean: 345 ± 30 g C m$^{-2}$ yr$^{-1}$; n = 22) reported for old-growth tropical forests

(Chave et al., 2010; Holland et al., 2015; Lamberty-Bond and Tompson, 2014). High litter

input, together with high annual temperature, can be another major factor explaining the

comparatively high soil $CO_2$ efflux rate in this southern conifer forest. This is somewhat

surprising as one would assume that organic matter mineralisation and thus soil $CO_2$ efflux is

reduced given the slow decomposition rate of kauri litter. In four kauri forests ranging from

pole to mature forests mean residence times between 9 and 78 years were estimated for 8 to

46 cm thick organic layers (Silvester and Orchard, 1999). According to Silvester and Orchard

(1999), sites with higher litter fall were accompanied by faster breakdown and no relationship

was found between litterfall and the depth of the organic layer. The organic layer in our study

sites was only 5 to 15 cm thick. Possible reasons for a lack of litter accumulation and build-

up of a thick organic layer are: removal and disturbance of the organic layer as a consequence

of tree fall and removal of five large kauri trees in the 1950s (Thomas and Ogden, 1983) and

stand age. Studies found that the proportion of lignin in litterfall increases in old-growth

stands and the change in the chemical composition of the litter layer coincides with the higher

content of twigs and reproductive structures in older forests (Gleixner et al., 2009). The

higher amounts of less degradable input in old-growth forests may lead to higher

accumulation rates (Gleixner et al., 2009). Reduced organic layer thickness can also be

explained by the topography of the study site (moderately to steep slope) as organic layer and

soil thickness have been found to decrease with steeper slope angles (Quideau, 2002).

While mean annual soil temperature partly explains the overall high mean soil $CO_2$ efflux

measured in this forest, soil temperature was not a very good predictor of the temporal

variation in soil surface $CO_2$ efflux. Independent of the regression model used, soil

temperature explained a small share (< 40%, Table 3) of the seasonal variation in soil surface

$CO_2$ efflux. This value is lower than the values reported for temperate forest ecosystems in

the Northern Hemisphere (Sulzman et al., 2005; Ngao et al., 2012; Bond-Lamberty and

Tompson, 2014). The poorer correlation was partly a function of small temporal differences

in soil temperature (< 5°C) compared to other temperate forests with a larger seasonal soil

temperature amplitude (> 10°C) (Paul et al., 2004).



Volumetric soil water content explained less than 18% of the temporal variability in soil
surface $CO_2$ efflux (Table 3). When SWC exceeded 40% a negative relationship between soil
surface $CO_2$ efflux and SWC was found. Excess SWC may negatively affect $CO_2$ efflux rates
by reducing soil aeration and thus $CO_2$ diffusivity (Janssens and Pilegaard, 2003). Further,
low levels of oxygen as result of high SWC decreases activity of plant roots (Adachi et al.,

2006) and the heterotrophic decomposition of soil organic matter (Linn and Doran, 1984).
This may be particularly relevant in the clayey soils under study.

## 4.2   Forest structure and the spatial variation in soil $CO_2$ efflux

The spatial variability (CV = 43%) of soil surface $CO_2$ efflux in this study is slightly higher

compared to other studies with similar numbers of measurements and/or plot size (32-39%;
Epron et al., 2006; Kosugi et al., 2007; Brechet et al., 2011). The higher spatial variation
might be related to differences in tree size and distribution across the plot. The stand is
clearly dominated by kauri trees in all size classes (Fig. 1). However, kauri occurs in clusters
around the four largest kauri individuals whose neighbourhood is generally characterised by

relatively few trees (see lower centre of Fig. 1). The influence of forest structure (here: kauri
tree distribution and tree size, $I_c$) on soil $CO_2$ efflux is confirmed by the significant
relationships between $I_c$ and soil $CO_2$ efflux, root biomass and mineral soil C:N ratio.
Previous studies have shown that kauri has significant effects on soil processes such as pH
and nitrogen cycling (Silvester 2000; Jongkind et al. 2007; Verkaik et al. 2007; Wyse et al.,

2014). This is the the first study showing that kauri exerts a substantial influence on soil C
related processes. Our results also corroborate a study by Katayama et al. (2009) suggesting
that the spatial arrangement of emergent trees in a tropical forest is an important factor for
generating spatial variation of soil $CO_2$ efflux. Studies in European beech forests also shown
that the combination of root, soil and stand structure help to understand the mechanisms

underlying soil $CO_2$ efflux and that forest structure has some influence on the spatial
variability of soil $CO_2$ efflux (Søe and Buchmann, 2005; Ngao et al., 2012).

The relationship between soil $CO_2$ efflux and forest structure was strongest within a radius of
5 m (Fig. 4.1a,b). In a tropical forest, the strongest correlation between soil $CO_2$ efflux and
forest structural parameters was within 6 m from the measurement points (Katayama et al.,

2009). A radius of 5 m also provided the best correlation between root biomass and $I_c$. As
measurements of the lateral root extension are not available for kauri, it remains unknown if



this distance equals the maximum lateral extension of fine roots from the trunk or represents the distance where fine root density is highest. Based on observations, large lateral roots of mature kauri trees often extent beyond the width of the crown and an extensive network of

fine roots extends from the lateral roots into the interface between organic layer and the mineral soil (Bergin and Steward, 2004). The radial fine root spread in mature Northern Hemisphere conifer stands varies considerably (6 - 20 m) depending on site characteristics and stand structure (Stone and Kalisz, 1991).

In contrast to other studies (e.g. Brechet et al., 2011; Katayama et al., 2009), we did not find a

significant correlation between litterfall and forest structure. Tree size and architecture have been reported to affect the pattern of litterfall distribution on the forest floor (Ferrari and Sugita 1996; Staelens et al., 2004; Zalamea et al., 2012). However, despite a 3-fold difference in tree size across the plot we did not see a significant effect of tree size on total litterfall. This is also reflected in a small within-plot variation in litterfall (CV = 20%, Table 1). This is

confirmed by a litterfall study in four remnant kauri forests where a small variation in litterfall (CV = 17 - 26%) was found across a wide range of litter trap positions (Silvester and Orchard, 1999).

Spatial variability in soil $CO_2$ efflux was largely attributed to soil temperature and the amount of fine root biomass and associated rhizosphere, with 97% of the variation explained. This

implies a relationship with tree productivity which is in agreement with findings from other conifer forests (Janssens et al., 2001; Lou and Zhou 2006). Although roots accounted for less then 30% of total $CO_2$ efflux recent research has shown that both recent photosynthate and fine root turnover can be important sources of C for forest soil $CO_2$ efflux (Epron et al., 2011; Warren et al., 2012) as discussed below.


## 4.3 Components of total soil $CO_2$ efflux

Collar insertion through the organic layer into the mineral soil resulted in a 17% reduction in soil $CO_2$ efflux. Similar reductions were found in other ecosystems and demonstrates that collar insertion by only a few centimetres cuts off fine roots (Heinemeyer et al., 2011) and

contributions by ectomycorrhizal fungal mats (Phillips et al., 2012) reducing total soil respiration. Thus, collar insertion can cause underestimation of total $CO_2$ efflux. This may be a particular problem in ecosystems where large amount of roots and mycorrhiza are found in



the organic layer and at the interface between the organic layer and an organic rich mineral soil as in this kauri forest.

The partitioning of total soil $CO_2$ efflux into its main components: heterotrophic respiration (oxidation of soil organic matter) and autotrophic respiration (root and associated mycorrhiza respiration) remains technically challenging. Differences in the proportion of autotrophic or heterotrophic respiration to total soil $CO_2$ efflux might vary not only among species and ecosystems but also with the method used for partitioning total soil $CO_2$ efflux (Kuzyakov,

2006; Subke et al., 2006; Millard et al., 2010). However, both techniques used in this study, trenching and regression-analysis, showed similar results. The proportion of autotrophic respiration in this kauri was between 25% (trenching) and 28% (regression analysis) of total soil surface $CO_2$ efflux. The contribution of autotrophic respiration to total soil $CO_2$ efflux can account for as little as 10% to more than 90% worldwide (Hanson et al., 2000) but values

of 45-50% are typical (Subke et al., 2006). Our estimate is at the lower end of values observed for Northern Hemisphere conifer and tropical broadleaf forests (30-70%, Epron et al., 2001; Högberg et al., 2001; Bond-Lamberty and Tompson, 2014; Taylor et al., 2015). This suggests that root/rhizosphere activity in this forest is comparatively low. However, a similar proportion of autotrophic respiration (23%) was estimated for a New Zealand old-

growth beech forest (Tate et al., 1993) and an old-growth Douglas-fir site in the Cascades, Oregon (23%) (Sulzman et al., 2005). Another factor accounting for the differences in values is the depth of trenching (Hansen et al., 2000; Kuzyakov, 2006; Bond-Lamberty et al., 2011). The contribution of autotrophic respiration may have been underestimated as we only trenched to 30 cm depth. It is recommended to trench to a depth beyond the main rooting

zone (Subke et al., 2006) and in some studies the trenched plots are dug down to the solid bedrock (Díaz-Pinés et al., 2010).

Total soil $CO_2$ efflux is not only directly affected by the amount of autotrophic respiration but also by the supply of C through root turnover and root exudates. The decomposition of root debris has been shown to increase microbial activity and thus heterotrophic respiration

(Göttlicher et al., 2006). Despite a low root/rhizosphere activity the total soil $CO_2$ efflux in a mycorrhizally-associated Douglas-fir forest was dominated by belowground contributions due to the large pool of rhizospheric litter with a relatively high turnover rate (Sulzman et al., 2005). In addition, root exudates containing carbohydrates, sugars and amino acids supply energy for the decomposition of soil C ('priming') (Högberg et al., 2001). Further, a recent



study showed that a common root exudate, oxalic acid, promotes soil C loss by releasing organic compounds from mineral-protected aggregates. This indirect mechanism has been found to result in higher C losses compared to simply increasing the supply of energetically more favourable substrates (Keiluweit et al., 2015).

Root activity may also affect physical soil conditions. In some studies, SWC and fine root

biomass were negatively correlated (Coomes and Grubb, 2000; Ammer and Wagner, 2002). High uptake of water by kauri fine roots concentrated in the organic layer may lead to lower SWC and slightly higher soil temperatures (Verkaik et al., 2007; Verkaik and Braakhekke, 2007). The drier conditions at the base of trees might be an indicator of good soil aeration that enhances the diffusivity of soil $CO_2$ into the air (de Jong and Schapper, 1972; Tang et al.,

565    2003).

The soil temperature – soil $CO_2$ efflux relationship was stronger for the inserted and trenched locations (= heterotrophic respiration) (Table 3). This is in line with other studies and suggests a higher sensitivity of heterotrophic respiration to temperature than autotrophic respiration (Kirschbaum, 1995; Boone et al., 1998). Although not significant, autotrophic

respiration tended to be lower during the dry summer 2013 compared to winter. A decrease in autotrophic respiration with drought have been reported for temperatue and tropical forests (Zang et al., 2014; Brunner et al., 2015; Doughty et al., 2015). This is in contrast to other studies which reported that dry conditions enhanced the growth of fine roots in the surface soil resulting in higher proportions of autotrophic respiration (Bhupinderpal-Singh et al., 2003;

Noguchi et al., 2007).

## 5   Conclusion

This is the first study quantifying the amount of soil $CO_2$ efflux in an old-growth kauri forest. Our findings suggest that the loss of soil $CO_2$ ($1315 \pm 77$ g C m$^{-2}$ yr$^{-1}$) from this forest type is

considerable. Although the contribution of autotrophic respiration is comparatively low ($<$ 30%), root biomass explained a high proportion of the spatial variation in soil $CO_2$ efflux. This suggests that, the total soil $CO_2$ efflux in this forest in not only directly affected by the amount of autotrophic respiration but also by the supply of C through roots and mycorrhiza. Any modification in root/rhizosphere will most likely result in long-term modifications of the

soil $CO_2$ efflux. This is of relevance given that many kauri forests are threatened by *Phytophthora agathidicida* (Weir et al., 2015) which infects the roots and can lead to tree



death (Than et al., 2013). This study is also the first to confirm that kauri not only exerts a strong control on soil pH and nitrogen cycling but also on soil carbon related processes. Aspects of the species and tree size distribution control of soil $CO_2$ efflux highlighted in this study demonstrates the need to include these parameters for better prediction of the spatial variability in soil $CO_2$ efflux.





**Data availability**

The data will be made available through figshare.


**Acknowledgements**

We thank Andrew Wheeler for his assistance in installing the soil $CO_2$ efflux chambers,
setting up the trenching experiment, measuring soil $CO_2$ efflux and developing and R script
for calculating soil $CO_2$ efflux; Roland Lafaele-Pereira and Chris Goodwin for assisting with
root sampling and sorting; Tristan Webb for helping with the soil $CO_2$ efflux measurements;
Hasinur Rahman for analysing the soil samples and Lena Weissert for running the regression
analysis in Matlab. This research was funded by a Faculty Research Development Fund grant
from the Faculty of Science, University of Auckland to LS and CMN.



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





Table 1. Descriptive statistics for litter, root, and soil characteristics. Samples were taken in the vicinity of the surface soil $CO_2$ efflux measurement locations (n = 12, except for root biomass, n = 10)

| Parameter | mean | STDEV | SE | median | min-max | CV % |
|---|---|---|---|---|---|---|
| Litterfall, ΣAug 12-Jan 14 (kg m$^{-2}$) | 1.9 | 0.4 | 0.1 | 2.0 | 1.1-2.2 | 20.2 |
| | | | | | | |
| Organic layer | | | | | | |
| Thickness (cm) | 8.8 | 2.3 | 0.9 | 8.2 | 6.2-12.2 | 26.1 |
| Root biomass (kg m$^{-2}$) | 0.8 | 0.9 | 0.3 | 0.3 | 0.02-2.7 | 115.6 |
| pH | 4.85 | 0.57 | 0.23 | 5.06 | 3.88-5.51 | 11.8 |
| C/N ratio | 43.9 | 10.4 | 4.2 | 43.2 | 31.4-58.7 | 23.7 |
| Carbon stock (kg m$^{-2}$) | 18.7 | 7.7 | 3.1 | 18.4 | 7.9-28.9 | 41.2 |
| Nitrogen stock (kg m$^{-2}$) | 0.45 | 0.18 | 0.07 | 0.45 | 0.22-0.77 | 40.0 |
| | | | | | | |
| Mineral soil | | | | | | |
| Root biomass, 0-15 cm (kg m$^{-2}$) | 2.2 | 1.6 | 0.5 | 1.6 | 0.7-6.3 | 93.8 |
| Root biomass, 15-30 cm (kg m$^{-2}$) | 0.7 | 1.2 | 0.4 | 0.4 | 0.2-3.9 | 97.7 |
| pH, 0-10 cm | 4.68 | 0.52 | 0.21 | 4.91 | 3.75-5.13 | 11.1 |
| C/N ratio, 0-10 cm | 16.1 | 1.9 | 0.8 | 16.2 | 13.7-19 | 12.1 |
| Carbon stock, 0-10 cm (kg m$^{-2}$) | 8.4 | 1.9 | 0.8 | 8.6 | 6.0-10.7 | 22.7 |
| Nitrogen stock, 0-10 cm (kg m$^{-2}$) | 0.53 | 0.13 | 0.05 | 0.52 | 0.40-0.75 | 24.1 |
| | | | | | | |
| Soil temperature (°C) | 14.2 | 0.2 | 0.1 | 14.2 | 14.0-14.5 | 1.4 |
| Volumetric soil water content (%) | 43.9 | 2.1 | 0.9 | 44.3 | 41.2-46.1 | 4.9 |




Table 2. Descriptive statistics of soil $CO_2$ efflux, soil temperature and volumetric soil water content across treatments and sampling sites. Measurements were conducted between August 2012 and Janury 2014. Different letters after the mean value for a given variable indicates a significant difference. Samples were separated into plot and trench for the statistical analysis due to different sampling
designs.

| Site/ Treatment | N | n | Soil $CO_2$ efflux ($\mu$mol $CO_2$ m$^{-2}$ s$^{-1}$) | | | | | Soil temperature (°C) | | | | | Volumetric soil water content (%) | | | | |
|---|---|---|---|---|---|---|---|---|---|---|---|---|---|---|---|---|---|
| | | | mean | STD SE | Med | Min Max | CV % | mean | STD SE | Med | Min Max | CV % | mean | STD SE | Med | Min Max | CV % |
| **Plot** | | | | | | | | | | | | | | | | | |
| Plot_Surface | 12 | 30 | 3.61a | 1.54 0.09 | 3.37 | 0.65 9.96 | 42.6 | 14.2a | 1.93 0.11 | 14.4 | 10.9 17.5 | 13.5 | 43.1a | 11.7 0.65 | 44.7 | 15.2 66.6 | 27.1 |
| Plot_Inserted | 12 | 30 | 2.98b | 1.30 0.07 | 2.72 | 0.69 8.02 | 43.6 | 14.1a | 1.94 0.10 | 14.1 | 10.9 17.4 | 13.8 | 44.7a | 10.3 0.56 | 46.6 | 15.2 62.3 | 23.0 |
| **Trench** | | | | | | | | | | | | | | | | | |
| Outsite_ Trench_Surface | 6 | 17 | 3.11x | 1.34 0.14 | 2.92 | 0.55 6.92 | 43.0 | 13.1x | 1.64 0.17 | 13.2 | 10.2 17.2 | 12.5 | 44.0x | 11.1 1.27 | 44.2 | 17.4 72.5 | 25.2 |
| Outside_Trench_ Inserted | 6 | 17 | 2.58y | 1.22 0.09 | 2.28 | 0.74 6.29 | 47.3 | 13.2x | 1.72 0.13 | 13.1 | 10.2 17.0 | 13.0 | 48.1y | 10.2 0.82 | 48.0 | 21.6 77.3 | 21.2 |
| Trench_Inserted | 6 | 17 | 2.34y | 0.96 0.08 | 2.14 | 0.67 5.30 | 41.0 | 12.9x | 1.70 0.14 | 13.0 | 10.1 16.9 | 13.1 | 56.8z | 8.4 0.74 | 56.4 | 20.2 76.5 | 14.8 |

N = number of locations per site, n = number of sampling dates between August 2012 and January 2014, Med = median




Table 3. Comparision of univariate soil temperature (T) or volumetric soil water content (W) only models and bivariate T-W models for the different treatments.

| Model | Var | Surface R² | Adj R² | RMSE | # | DFE | Inserted R² | Adj R² | RMSE | # | DEF | Trenched+Inserted R² | Adj R² | RMSE | # | DEF |
|---|---|---|---|---|---|---|---|---|---|---|---|---|---|---|---|---|
| **Plots** | | | | | | | | | | | | | | | | |
| Linear | T | 0.331 | 0.308 | 0.640 | 2 | 28 | 0.569 | 0.554 | 0.473 | 2 | 28 | | | | | |
| Lloyd and Taylor | T | 0.000 | -0.074 | 0.797 | 3 | 27 | 0.567 | 0.534 | 0.483 | 3 | 27 | | | | | |
| Logistic | T | 0.406 | 0.362 | 0.614 | 3 | 27 | 0.569 | 0.537 | 0.482 | 3 | 27 | | | | | |
| Q10 model | T | 0.401 | 0.357 | 0.617 | 3 | 27 | 0.552 | 0.519 | 0.491 | 3 | 27 | | | | | |
| Quadratic | T | 0.418 | 0.375 | 0.608 | 3 | 27 | 0.567 | 0.534 | 0.483 | 3 | 27 | | | | | |
| Linear | W | 0.036 | 0.000 | 0.756 | 2 | 28 | 0.489 | 0.470 | 0.525 | 2 | 28 | | | | | |
| Quadratic | W | 0.178 | 0.115 | 0.711 | 3 | 27 | 0.510 | 0.472 | 0.523 | 3 | 27 | | | | | |
| Polynomial | T,W | 0.537 | 0.501 | 6.409 | 3 | 26 | 0.589 | 0.557 | 5.571 | 3 | 26 | | | | | |
| Q10 Hyperbolic | T,W | 0.585 | 0.535 | 6.185 | 4 | 25 | 0.584 | 0.534 | 5.711 | 4 | 25 | | | | | |
| | | | | | | | | | | | | | | | | |
| **Trench** | | | | | | | | | | | | | | | | |
| Linear | T | 0.000 | -0.067 | 0.899 | 2 | 15 | 0.206 | 0.153 | 0.323 | 2 | 15 | 0.233 | 0.182 | 0.296 | 2 | 15 |
| Lloyd and Taylor | T | 0.000 | -0.143 | 0.931 | 3 | 14 | 0.003 | -0.139 | 0.375 | 3 | 14 | 0.271 | 0.167 | 0.299 | 3 | 14 |
| Logistic | T | 0.019 | -0.121 | 0.922 | 3 | 14 | 0.196 | 0.081 | 0.337 | 3 | 14 | 0.271 | 0.167 | 0.299 | 3 | 14 |
| Q10 model | T | 0.077 | -0.055 | 0.894 | 3 | 14 | 0.208 | 0.095 | 0.334 | 3 | 14 | 0.233 | 0.123 | 0.307 | 3 | 14 |
| Quadratic | T | 0.149 | 0.027 | 0.859 | 3 | 14 | 0.208 | 0.095 | 0.334 | 3 | 14 | 0.254 | 0.147 | 0.303 | 3 | 14 |
| Linear | W | 0.023 | -0.052 | 0.875 | 2 | 15 | 0.146 | 0.085 | 0.347 | 2 | 15 | 0.063 | -0.003 | 0.330 | 2 | 15 |
| Quadratic | W | 0.115 | -0.033 | 0.867 | 3 | 14 | 0.148 | 0.017 | 0.360 | 3 | 14 | 0.096 | -0.043 | 0.336 | 3 | 14 |
| Polynomial | T,W | 0.376 | 0.272 | 8.864 | 3 | 12 | 0.333 | 0.231 | 8.603 | 3 | 13 | 0.063 | -0.081 | 6.189 | 3 | 13 |
| Q10 Hyperbolic | T,W | 0.392 | 0.226 | 9.140 | 4 | 11 | 0.333 | 0.167 | 8.955 | 4 | 12 | 0.103 | -0.122 | 6.305 | 4 | 12 |

$R^2$, adjusted $R^2$ = coefffient of determination; RMSE = root mean aquare error, DFE = Degrees of Freedom for Error; # = numer of fitted parameters; y = soil $CO_2$ efflux; x = soil temperature; z = volumetric soil water content, Equations: Linear T, W: $y = a*x + b$; Lloyd and Taylor T: $y = a*exp(-b/(x+273.16+c))$; Logistic T: $y = a/(1+exp(b*(c-x)))$; Q10 model T: $y = a*b^{(x-10)}/10+c$; Quadratic T, W: $y = a*x^2 + b*x + c$; Polynomial T, W: $y = a + bx + cz$; Q10 Hyperbolic T, W: $y = (b^{(x-10)}/10)*((a+z*c+d/z))$



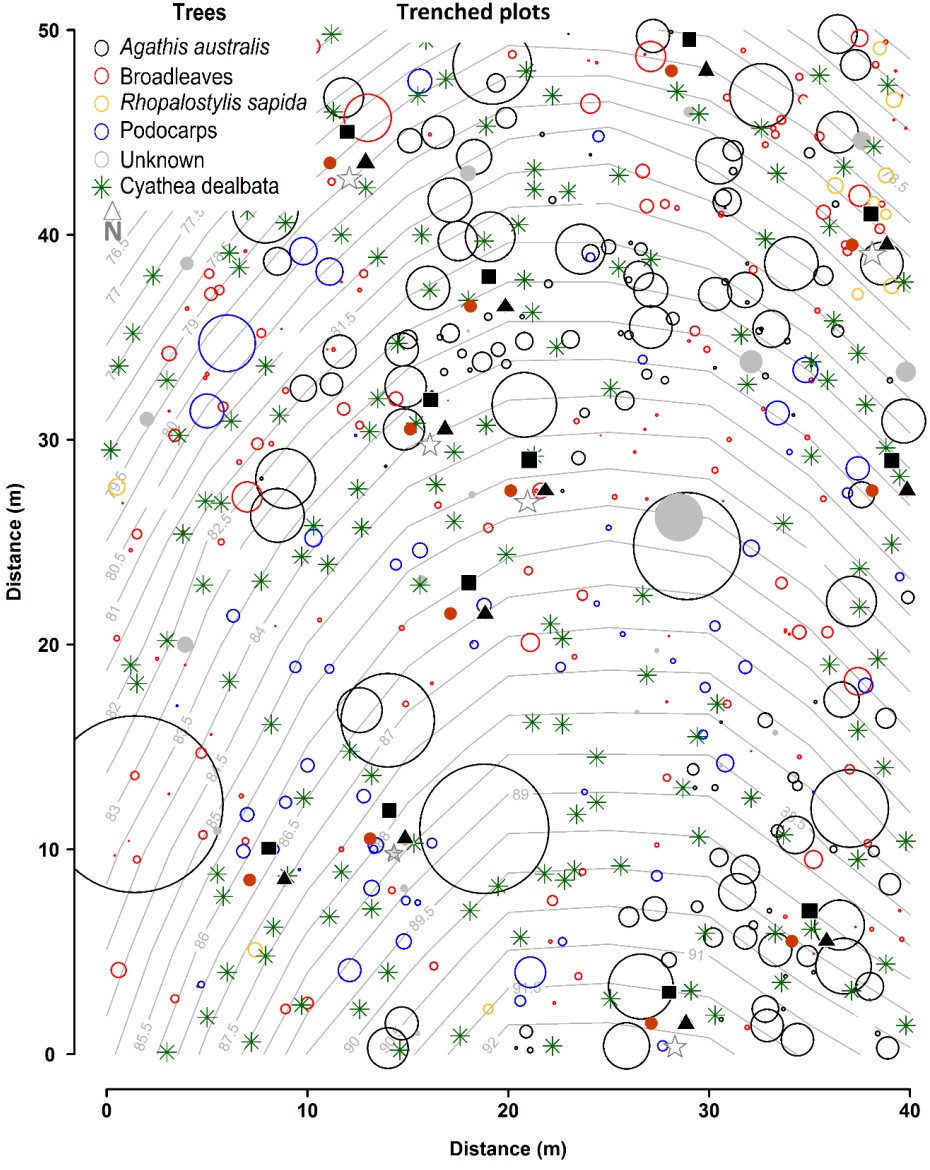

Figure 1. Overview of the research plot showing the position of all trees ≥ 2.5 cm diameter (larger circles represent larger diameter at breast height), surface soil $CO_2$ efflux locations (black filled square), inserted collars (clusters of three, red filled circle), litter traps (black filled triangle), root mass sampling locations (grey open stars).






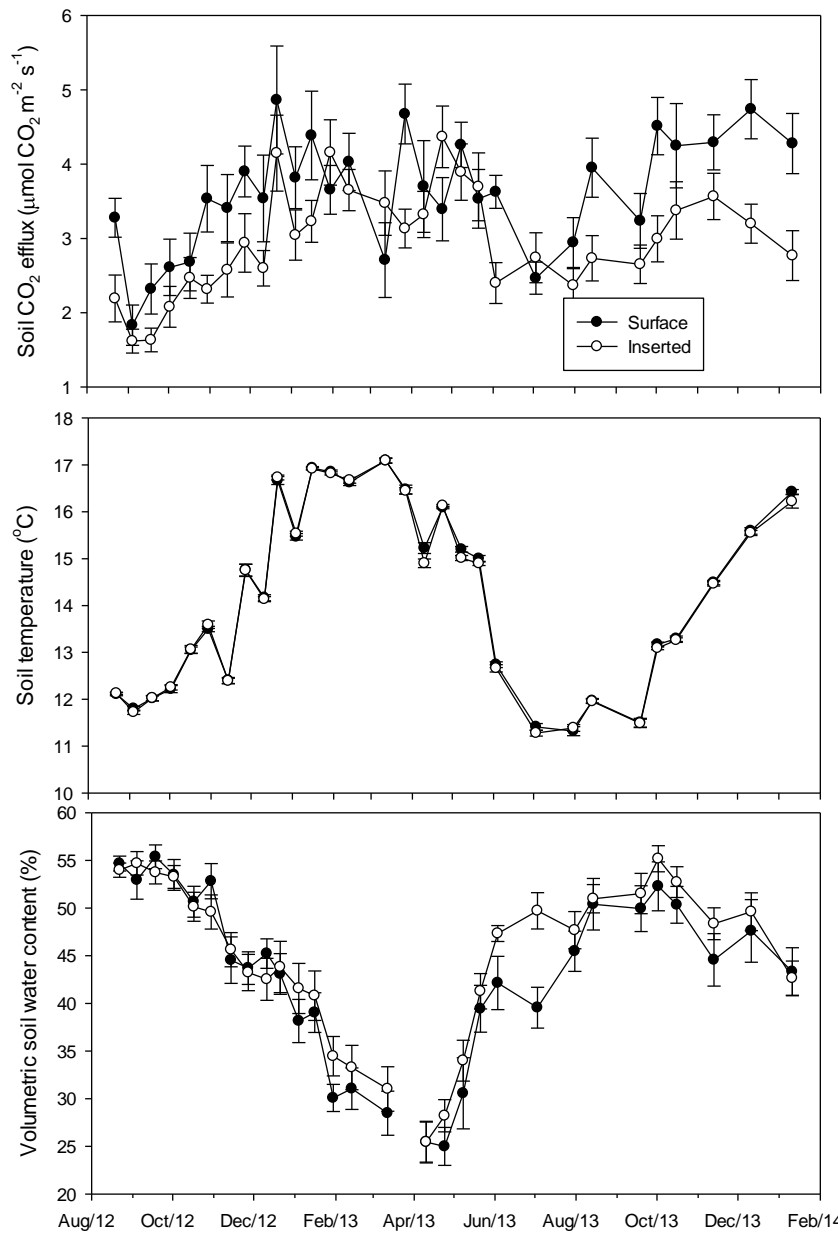

Figure 2. Soil CO$_2$ efflux (A), soil temperature (B) and volumetric soil water content (C)
measured in the research plot from August 2012 to January 2014. Values show mean ± SE of
Plot_Surface and Plot_Inserted collars (n = 12). Volumetric soil water content was not
measured in March 2013 due to equipment failure.



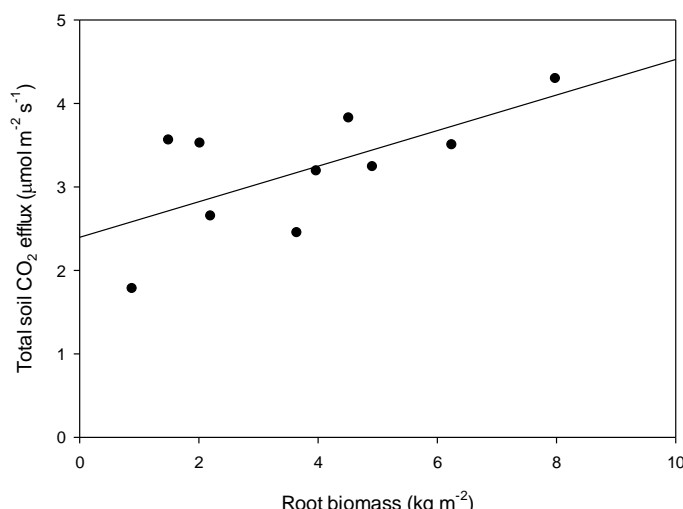

Figure 3. Regression of total root biomass to 30 cm depth vs total soil $CO_2$ efflux. Surface (=
total) soil $CO_2$ efflux = 0.213 x root biomass + 2.49 ($R^2$ = 0.394, p = 0.042).




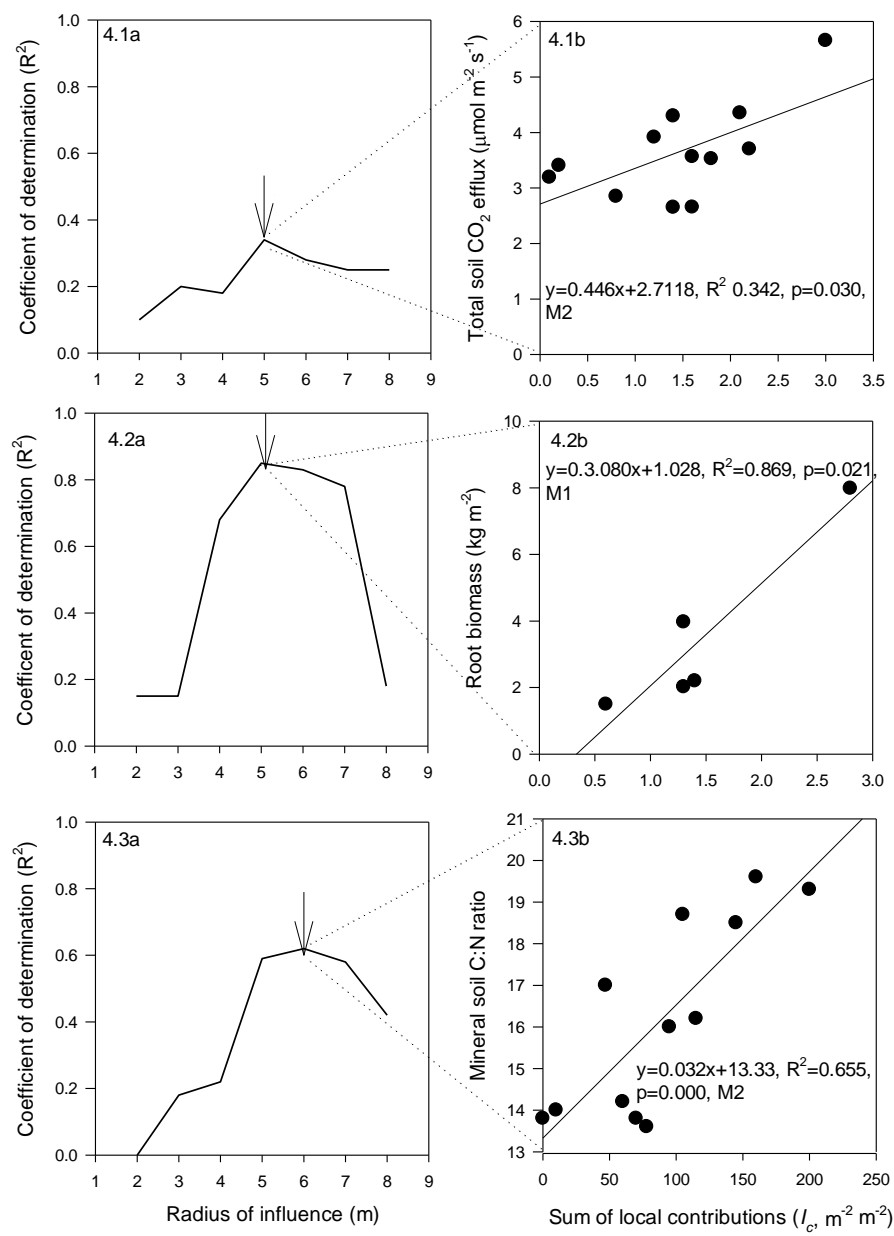

Figure 4. Relationships between the sum of local contribution indices of surrounding trees within the fitted radius of influence and soil $CO_2$ efflux (4.1.a,b), root biomass (4.2.a,b) and mineral soil CN ratio (4.3a,b). The arrows in panel a indicate the best coefficients of variation (highest $R^2$ value) with models shown in panel b. M1 = univariate model, $I_c$ = S), M2 = linear model, $I_c$ = S x (1-d/r where S = trunk cross section area (S, in cm²), d = distance between the trees and the measurement point (d, in m), a = coefficient of form, r = fitted radius of influence (r, in m).
