# Peer review of "Soil CO2 efflux in an old-growth southern conifer forest (*Agathis australis*) – magnitude, components, and controls"

_SOIL, 2016_

## Referee Comment (RC1) · Anonymous Referee #1 · 17 May 2016

Review of manuscript soil-2016-21: Soil CO2 efflux in an old-growth southern conifer forests (Agathis australis) – magnitude, components, and controls

=== General comments

The manuscript describes a study of soil respiration in a native forest of New Zealand. It is well written, meaning correct and fluent language, a clear introduction and presentation of the methods and results. It manages to describe well the characteristics of soil CO2 efflux in this type of forest and has the advantage of being the first such study in this particular ecosystem. The originality of the study is mostly if not entirely the

result of this last point. While it presents correlation analyses and finds temperature and root biomass as the most important factors explaining the CO2 fluxes, it remains otherwise mostly descriptive. Some results, interpretations and conclusions are not entirely convincing. In particular, I would question the correctness of the model fitting section.

=== Specific comments

Introduction and Methods are well written. Here I find nothing to question. When trenching or inserting deep collars, severed roots can add to the decomposing pool and change the estimate of heterotrophic respiration. How were decomposing roots accounted for in this study?

In section 3.3 you describe fitting models for the T response but fail to mention the most common used i.e. Q10 or LT, etc.

The Q10-function is usually equivalent to an exponential function and has only 2 parameters, i.e. a * Q10^((T-Tref)/10). Why do you have 3 parameters for the Q10 function? Is one a constant? Please check your functions in Table 3. Everything in the exponent should be closed by parenthesis. Also, the fact that you improved your R2 with a bivariate model but have much larger RMSE is not consistent. Check that your calculations are correct. Adding explanatory variables should only reduce the RMSE if you are using the same data.

It would be good to have plots showing the response to T and M.

In the discussion you calculate an average and compare with other ecosystems. Using the average of your measurements is incorrect. Since there is a T and root effect you should account for these when getting yearly estimates. At least use the T relationship, since T at night is probably lower, so the yearly average is lower than that of your measurements.

You discuss how the vegetation may control the amount of CO2 efflux. The question of

whether your system is near equilibrium is important here. If a forest is near equilibrium, the quality of the litter is important only in determining the stock sizes, not the $CO_2$ fluxes. The latter will be equal to the amounts of input.

When discussing the effect of T, make clear that your T range is small, which does not mean there is little T effect, just that you cannot see it. In terms of the average yearly T, this will probably have a larger effect in how it affects the productivity fo the vegetation, so indirectly through litter input.

In conclusions, you state that the study has found that the vegetation type exerts a strong influence on soil carbon related processes. This is an effect of all land vegetation and is no finding by itself, thus making for a very weak conclusion. An insight on the vegetation effect on the soil C stocks or some other more specific observation should come here. Also, you mention that species effects were should in the study, however no species comparison was made, so mentioning species effects is incorrect, here and in the abstract.

Lines 317-319 This line is not clear to me

---

## Referee Comment (RC2) · Anonymous Referee #2 · 17 May 2016

General remarks: The interesting manuscript (ms) represents the investigations within a native forest in New Zealand with the aim to characterize dependencies between forest structure, soil properties, meteorological conditions and soil respiration processes. The ms has clear objectives and represents a good contribution to scientific progress in the interdisciplinary field of abiotic and biotic soil respiration influences. In order to describe temporal variability the data interpretation is based on times series over 18 month of CO2 efflux measurements. Furthermore, the authors considered a huge number of relevant references, giving a comprehensive insight and the chance to compare the approach with the results from other researchers. The overall quality of the

manuscript is high. It offers interesting insights into the soil CO2 efflux within an old-growth kauri forest and the main controlling factors for such a forest site. The entire results are discussed based on sound statistical analysis. However, in my opinion there are some issues which need to be discussed more in detail to close some minor gaps improving the ms.

Specific comments:

Line 36: Janssens et al., 2001 – two times in reference list – which one is meant here? → 2001a, b

Line 138 and Figure1: The figure 1 is not easy to understand concerning the experimental set up. There are some items, which are not explained in the legend: filled grey circles, filled grey stars, grey lines (I supposed it is the topography in m a.s.l.?) Where are the trenched plots? There is only text at the upper edge.

In the context topography: you mention a potential dependency between topography and organic layer thickness (line 448-450). I would strongly recommend analyzing a functional trend between soil moisture and topography and hence, maybe some influences on soil respiration. From your data compilation it is not clear to see, but maybe the soil moisture differences you mentioned in Table 2 for the trenched plots could be superimposed by topographic driven soil moisture differences. You can find some discussions in recent papers, e.g.,

Masamichi Takahashi , Keizo Hirai , Pitayakon Limtong , Chaveevan Leaungvutivirog , Samreong Panuthai , Songtam Suksawang , Somchai Anusontpornperm & Dokrak Marod (2011) Topographic variation in heterotrophic and autotrophic soil respiration in a tropical seasonal forest in Thailand, Soil Science and Plant Nutrition, 57:3, 452-465, DOI: 10.1080/00380768.2011.589363

Wang, B., Zha, T.S., Jia, X., Gong, J.N., Wu, B., Bourque, C.P.A., Zhang, Y., Qin, S.G., Chen, G.P., Peltola, H., 2015. Microtopographic variation in soil respiration and its

controlling factors vary with plant phenophases in a desert–shrub ecosystem. Biogeosciences 12, 5705-5714.

Line 221: . . . plant material (45% C, 25 2.3% N) . . . → 25 ??

Line 312 and Figure 2: You mentioned a relation between high CO2 efflux and heavy rain events (as described and shown in paper Macinnis-Ng & Schwendenmann, 2015). Why do not show precipitation information in figure 2? I suppose, that graph would visually support very well your interpretation!

Line 333: . . .Outside_Trench_Insered . . . t is missing

Line 537: reference Epron et al., 2001 is not in reference list

Line 564: . . .de Jong and Schappert. . .

Line 957: January

Line 958: The different letters .. indicates a significant difference . . . between what? Mean and Median? What means a, b, x, y, z?

Line 965a: The determined regression coefficients are in all cases very weak – hence, it is not really a convincing correlation! As an example, you could include a figure to show the different modelled approaches.

Line 965b: Table 3 subscription . . . adjusted R2 = coefficient. . . RMSE = root mean aquare

Line 798: reference Metcalfe et al., 2011 is not mentioned in text

---

## Referee Comment (RC3) · Anonymous Referee #3 · 27 May 2016

This study reports measurements of soil respiration and their component parts in a kauri forest in New Zealand. Both trenching and statistical techniques are used to partition the total soil efflux into auto and heterotrophic respiration. Statistical methods are then used to investigate the temporal controls by environmental parameters. Tree root biomass and "tree influence" are used to look for controls on the spatial variability in soil CO2 efflux. There are extensive references and good comparisons with data from the NH. The interest in this manuscript doesn't so much lie in its respiration results and partitioning, but rather its combination of soil respiration and spatial patterns relating to root biomass.

[Figure]

An attempt was made to test soil respiration methodology with comparisons between surface and inserted rings. However, this was not well described in the introduction and I was confused to why they did this. In the Experimental Setup the reason for the inserted and surface chambers should be explained. I found the overall aims of the manuscript confusing and this wasn't helped by the description of the methods and the 5 different types of soil surface measurements. A better way to arrange this might be to describe each aim and then the methods that go with it. Eg Collar insertion depth, respiration partition, annual soil respiration, spatial variability vs temporal variability. It would be good to see the data, especially the relationship between soil respiration (total, Ra, Rh) and temperature.

The comparison of average soil respiration with other sites does not take temperature into account. It would be better to compare R10 or Q10 values. Alternatively you could use your know relation with temperature to adjust your values to the same temperature at other sites. The collars were inserted in November 2011 and efflux measurements commenced in January 2012. This does not leave enough time for the roots to decompose; therefore this is not truly a measurement of heterotrophic respiration. How was this accounted for and was there a decrease in RH over time as the roots decayed? In the study site description, it would be good to know that the forest had been disturbed by tree removal and may not be in equilibrium. L 187 states that efflux was measured on a number of days immediately after trenching, but this data are not presented. L 203 Where was this temperature measured, in the chamber? L 221 delete "of", also (45% C, 25 2.3% N) doesn't make sense. L 236 You state that there are two replicates, but on L 182 it says "one location" L 255 spelling of "Surfave" L 263 Table 3 is referenced before Table 2 L 294 Please state which subplot is being refeered to (Fig 2.?), also what is 14.2 +/- 0.1 a SD of a SEM L 294 Please refer to the months as well as the season L 300 Fig 2A L 303 Change "locations" to times. L 313 I couldn't see an increase in variability during the dry summer of 2013 (I gather this is Jan – Mar 2013?) L 317 The data for summer/early autumn 2012 is not presented. L 329 Was SWC really affected by collar insertion, if so how? L 367 I am not sure how we can see in the table

that there is a sign changes around 40% soil water content. L 377 Change Table 1 to Table 2 L 381 Table 1 does not show the 0-30 cm values L 399 3.47 umols is 3.6 umol on line 303 L 458 Fig 2 the temperature difference is > 5 degrees" but Table 2 shows 6.6 degrees). Maybe there is a temperature response but it doesn't show up with such a small temperature range. L 466 This sentence doesn't make sense. L 482 Are the mature kauri at the site emergent? If so then state this in the site description. L 516 State early on that you are going to test the effect of collar depth on effluxes.

Table 1 It would be good to have the litterfall summed over a year so it can be compared with other sites. Table 2 I don't think it is necessary to show both the STD and the SE. Outside is misspelt as "Outsite_Trench_Surface". Using x, y and z for significant differences is confusing, stick with "c, d, e". Table 3. Why are some numbers italicized? You need to define what a, b and c are.

Fig 1 "Unknown"? – seems like it should be possible to get an identification of the tree species over the two years of the study, surely it can be classified as a broadleaf or not. The unknowns are filled circles, but open circles in the legend. There are two types of stars. How much are the size of the circles are scaled by the diameter? The 0.5 m contour lines are not really needed unless referred to in the manuscript. The plot has "trenched plots" as a title, this should be removed. Fig 2. These plots need to be labelled a, b, c. The sample points are joined up with lines; I cannot see why this can't be done in the SWC graph. The figure caption should indicate that these are means and state which soil efflux is being referred to (surface, trenched, inserted, plot or outside). Fig 4. Subplots should be labelled a-f. On subplot 4.2b the equation is wrong and ends up outside the 2nd x axis. L 984 is missing a ")", S is given in cm2 but the units along the x axis are in m2. Is m-2 m-2 correct? A is described as coefficient form, but does not appear in the equation.

---

## Author Comment (AC1) · 28 Jun 2016

Referee #1 We thank the referee for the constructive comments which helped to improve the quality of the paper. Please find below a detailed response to the each of the comments.

General comments The manuscript describes a study of soil respiration in a native forest of New Zealand. It is well written, meaning correct and fluent language, a clear introduction and presentation of the methods and results. It manages to describe well the characteristics of soil CO2 efflux in this type of forest and has the advantage of

being the first such study in this particular ecosystem. The originality of the study is mostly if not entirely the result of this last point. While it presents correlation analyses and finds temperature and root biomass as the most important factors explaining the CO2 fluxes, it remains otherwise mostly descriptive. Some results, interpretations and conclusions are not entirely convincing. In particular, I would question the correctness of the model fitting section.

Response: We re-analysed the data set using the Q10 and modified Arrhenius (Lloyd and Taylor 1994) function to test the temperature response of total soil CO2 efflux and heterotrophic (for details see response 2 regarding section 3.3)

Specific comments 1. Introduction and Methods are well written. Here I find nothing to question. When trenching or inserting deep collars, severed roots can add to the decomposing pool and change the estimate of heterotrophic respiration. How were decomposing roots accounted for in this study?

Response: We did not correct our estimate of soil CO2 efflux for decomposing root-derived CO2 flux. We did not observe a significant insertion/trenching related change in heterotrophic respiration. We don't have data on kauri root decomposition but previous studies showed that kauri litter is characterized by very long residence times (between 9 and 78 years, Silvester and Orchard, 1999). To address the effect of root decomposition-derived CO2 fluxes we included a statement in the methods section and modified the discussion as follows: "Cutting roots through inserting deep collars and trenching increases the dead root biomass (Heinemeyer et al., 2011). As we did not correct our estimates of soil CO2 efflux for decomposing root-derived CO2 fluxes the heterotrophic respiration may have been slightly overestimated (Hanson et al., 2000; Kuzyakov, 2006; Ngao et al., 2012)."

2. In section 3.3 you describe fitting models for the T response but fail to mention the most common used i.e. Q10 or LT, etc. The Q10-function is usually equivalent to an exponential function and has only 2 parameters, i.e. a * Q10ЁЕ((T-Tref)/10). Why

do you have 3 parameters for the Q10 function? Is one a constant? Please check your functions in Table 3. Everything in the exponent should be closed by parenthesis. Also, the fact that you improved your R2 with a bivariate model but have much larger RMSE is not consistent. Check that your calculations are correct. Adding explanatory variables should only reduce the RMSE if you are using the same data. It would be good to have plots showing the response to T and M.

Response: The Q10 function (Schlentner and van Cleve, 1985) we used is a sigmoid function with three parameters (a = lower limit of soil CO2 efflux, a+1/b = maximum flux, c = Q10 related parameter).

Based on the recommendation by referee 1 and referee 3 we re-analysed our data set using the most commonly used temperature response functions (linear, exponential Q10 and modified Arrhenius function). The linear temperature response function provided the best fit, explaining 44% of total soil CO2 efflux and 53% of the heterotrophic respiration (see Figure below). This new figure (Figure 4) will be included in the manuscript. The methods, results and discussion sections have been modified as follows:

Methods Section 2.5 Data analysis "Univariate and bivariate models were used to investigate the relationship between total soil CO2 efflux, heterotrophic and autotrophic respiration and the abiotic factors soil temperature and volumetric soil water content. Data from within the research plot and trench sampling points were combined. The temperature response of soil CO2 efflux was tested using a linear, exponential Q10 (van't Hoff 1898) and modified Arrhenius function (Lloyd and Taylor, 1994). Linear and hyperbolic functions were used to assess the soil water dependence of soil CO2 efflux. The combined effect of soil temperature and soil water content on soil CO2 efflux was tested using a polynomial function. Coefficient of determination (R2) and standard error of estimate (SEE) were used to evaluate model performance."

Results Section 3.3 "The linear temperature function explained around 44% of the temporal variation in total soil $CO_2$ efflux (Figure 4A, Table 3). Exponential ($R^2=0.13$) and modified Arrhenius ($R^2=0.17$) functions resulted in lower $R^2$ values (Table 3). The $Q_{10}$ values for total soil $CO_2$ efflux was 1.6 (Table 3). A slightly stronger soil temperature response was found for heterotrophic respiration (linear function, $R^2=0.530$, Figure 4B) with a $Q_{10}$ value of 2.2 (Table 3). No significant relationship was found between soil temperature and autotrophic respiration (Figure 3C). Neither a linear nor a quadratic function resulted in a significant relationship between SWC and total soil $CO_2$ efflux (Figure 4D). Heterotrophic respiration decreased significantly with increasing SWC ($R^2=0.590$, Figure 4E). In contrast a weak, but significant positive relationship was found between SWC and autotrophic respiration (Figure 4E). Bivariate polynomial functions did not result in higher $R^2$ values compared to univariate models (Table 3)."

Discussion Section 4.1: "While mean annual soil temperature partly explains the overall high mean soil $CO_2$ efflux measured in this forest, soil temperature was not a very good predictor of the temporal variation in total soil $CO_2$ efflux. Independent of the regression model used, soil temperature explained a small proportion (< 44%, Figure 4A, Table 3) of the seasonal variation in total soil $CO_2$ efflux. In temperate forest ecosystems in the Northern Hemisphere (Ngao et al., 2012; Bond-Lamberty and Tompson, 2014) soil temperature often explains more than 50% of the temporal variability in total soil $CO_2$ efflux. It is important to note that the soil temperature range in this kauri forest was narrow (around 7°C) compared to other temperate forests with a larger seasonal soil temperature amplitude (> 10°C, Paul et al., 2004). Thus, a seasonal temperature effect may not have been visible in this kauri forest. The $Q_{10}$ value (1.6, $R^2=0.172$, Table 3) was at the lower end of the range reported for mixed and evergreen forests ($Q_{10}\_10$-20°C; 0.5-5.6; Bond-Lamberty and Tompson, 2014). However, low $Q_{10}$ values have also been reported for other conifer forest, especially at sites characterized by mild winters (Borken et al. 2002; Curiel Yuste et al., 2004; Sulzman et al., 2005). Low $Q_{10}$ values in evergreen forests have been explained by the lack of a distinct seasonality in photosynthesis and substrate supply (Curiel Yuste et al., 2004)."

Figure 4 (see below).

3. In the discussion you calculate an average and compare with other ecosystems. Using the average of your measurements is incorrect. Since there is a T and root effect you should account for these when getting yearly estimates. At least use the T relationship, since T at night is probably lower, so the yearly average is lower than that of your measurements.

Response: The yearly estimate was re-calculated using the linear response function (best fit) and soil temperature data (5 cm depth, 30 min averages). The revised annual estimate resulted in a slightly higher annual estimate (1324 $\pm$ 121 g C m-2 yr-1).

4. You discuss how the vegetation may control the amount of CO2 efflux. The question of whether your system is near equilibrium is important here. If a forest is near equilibrium, the quality of the litter is important only in determining the stock sizes, not the CO2 fluxes. The latter will be equal to the amounts of input.

Response: The forest stand is dominated by a few emergent (up to 300 year-old) kauri trees. It is unlikely that this forest in near equilibrium as tree fall and removal of five large kauri trees in the 1950s created gaps which are now dominated by a cohort of younger kauri trees. The following statement has been added.

2.1 Study site "Kauri tree size distribution differs within the plot. Four emergent kauri trees (up to 180 cm in DBH, approx. 300 year-old) are found on the upper slope of the plot. At the lower slope tree fall and removal of five large kauri trees in the 1950s created gaps which are now dominated by a cohort of younger kauri trees."

5. When discussing the effect of T, make clear that your T range is small, which does not mean there is little T effect, just that you cannot see it. In terms of the average yearly T, this will probably have a larger effect in how it affects the productivity of the vegetation, so indirectly through litter input.

Response: The discussion on the effect of temperature on soil CO2 efflux has been
modified (see above, Response 2).

6. In conclusions, you state that the study has found that the vegetation type exerts a strong influence on soil carbon related processes. This is an effect of all land vegetation and is no finding by itself, thus making for a very weak conclusion. An insight on the vegetation effect on the soil C stocks or some other more specific observation should come here. Also, you mention that species effects were should in the study, however no species comparison was made, so mentioning species effects is incorrect, here and in the abstract.

Response: We found strong relationships between the "index of local contribution" (a measure of kauri tree size and distribution) and total soil $CO_2$ efflux, root biomass and mineral C:N ratio suggesting that the spatial arrangement of kauri trees influences soil characteristics and soil $CO_2$ efflux. This is in line with previous findings showing that kauri trees exert a strong control on soil pH and soil nitrogen (Silvester 2000; Jongkind et al. 2007; Verkaik et al. 2007; Wyse et al., 2014). In our conclusion we wanted to highlight the importance of investigating biotic factors (here: kauri tree distribution and size as a measure of forest structure) in soil carbon related studies.

The abstract and conclusion sections have been revised accordingly ("Our findings suggest that biotic factors such as tree structure should be investigated in soil carbon related studies.")

7. Lines 317-319 This line is not clear to me

Response: This statement has been deleted.
* * *
[Figure]

**Fig. 1.** Upper panel. Relationship between soil temperature and total soil CO2 efflux (A), heterotrophic respiration (B) and autotrophic respiration (C). Lower panel: Relationship between volumetric water cont

---

## Author Comment (AC3) · 28 Jun 2016

Referee #3 We highly appreciate the referee's comments and suggestions, which helped improve the quality of the manuscript. Please find below a detailed response to the each of the comments.

General comments

This study reports measurements of soil respiration and their component parts in a kauri forest in New Zealand. Both trenching and statistical techniques are used to partition the total soil efflux into auto and heterotrophic respiration. Statistical methods

are then used to investigate the temporal controls by environmental parameters. Tree root biomass and "tree influence" are used to look for controls on the spatial variability in soil CO2 efflux. There are extensive references and good comparisons with data from the NH. The interest in this manuscript doesn't so much lie in its respiration results and partitioning, but rather its combination of soil respiration and spatial patterns relating to root biomass.

Specific comments

1. An attempt was made to test soil respiration methodology with comparisons between surface and inserted rings. However, this was not well described in the introduction and I was confused to why they did this.

Response: We revised section 1 as follows: "The aim of this study was to determine the magnitude, components and the driving factors of soil CO2 efflux in an old-growth southern conifer forest. The specific objectives of our study were: (i) to quantify total soil CO2 efflux, (ii) to test the effect of collar depth on soil CO2 efflux, (iii) partition total soil CO2 efflux into autotrophic and heterotrophic respiration, (iv) to identify the factors controlling the temporal variation of total soil CO2 efflux and its component fluxes, and (v) to test the effect of kauri tree size and distribution on total soil CO2 efflux and soil properties. In order to achieve the objectives we measured soil CO2 efflux in an old-growth kauri stand over 18 months. We used direct (trenching) and indirect (regression technique) approaches to partition total soil CO2 efflux (organic layer plus mineral soil to 30 cm depth) into the autotrophic and heterotrophic components. Given that old-growth kauri forests are often characterised by thick organic layers we used deep collar insertion to assess the effect of insertion depth on soil CO2 efflux and to quantify to proportion of autotrophic and heterotrophic respiration in this layer."

2. In the Experimental Setup the reason for the inserted and surface chambers should be explained.

Response: We revised the methods section to improve the links between objectives

and the experimental set up as follows. We decided not to include the data from the "Outside_Trench_Inserted" sampling points as the results confirmed the findings in the plot (17% reduction in total CO2 efflux due to deep collar insertion).

2.2. Experimental set-up "To quantify the effect of insertion on total soil CO2 efflux and to determine the proportion of autotrophic and heterotrophic respiration to total soil CO2 efflux in the organic layer, a cluster of three 'deep' PVC collars (10 cm in diameter, 20 cm in height) was inserted next to each sampling point for surface soil CO2 efflux measurements. Three collars per cluster were spaced evenly around the circumference of a circle 2 m in diameter, with small adjustments in the spacing to accommodate large roots. Each collar was driven right through the organic layer and 1-2 cm into the mineral soil layer to cut off the roots growing in the organic layer. In order to prevent CO2 uptake, any vegetation inside the collars was regularly removed. The thickness of the organic layer at each grid point was measured using a ruler outside each collar. The 'deep' collars were inserted in November 2011 and left in place over the measurement period. Efflux was measured from August 2012 to January 2014. Here after, these sampling points are known as Plot_Inserted."

"We used the trenching approach to separate heterotroph and autotrophic respiration in the organic layer plus mineral soil to 30 cm depth. To avoid disturbing the long-term re-search plot the trenching experiment was set-up directly adjacent to the research plot. In July 2012, six 2 x 2 m plots were trenched to 30 cm depth based on a preliminary study showing that the majority of fine roots (over 80%) are located in the organic layer and top 30 cm of the mineral soil. The trenches were double-lined with a water perme-able polypropylene fabric and backfilled. During trenching, trampling and disturbance inside the 2 x 2 m plots were avoided as far as possible. Two types of measurements were conducted in the trenched plots. First, surface soil CO2 efflux was measured at two sampling points outside each trenched plot (Outside_Trench_Surface) in the same way as the Plot_Surface samples were measured (see above. Second, two collars were randomly placed inside the trenched plot (Trench_Inserted) and were inserted

Interactive
comment

1-2 cm into the mineral soil layer (deep collars) as described above. Soil $CO_2$ efflux was measured bi-weekly to monthly from August 2012 until December 2013."

3. I found the overall aims of the manuscript confusing and this wasn't helped by the description of the methods and the 5 different types of soil surface measurements. A better way to arrange this might be to describe each aim and then the methods that go with it. Eg Collar insertion depth, respiration partition, annual soil respiration, spatial variability vs temporal variability.

Response: We modified the methods section as suggested (see response 2). We decided not to include the data from the "Outside_Trench_Inserted" sampling points as the results confirmed the findings in the plot (17% reduction in total $CO_2$ efflux due to deep collar insertion).

4. It would be good to see the data, especially the relationship between soil respiration (total, Ra, Rh) and temperature.

Response: A figure showing the relationship between total soil $CO_2$ efflux, heterotrophic and autotrophic respiration has been included. We re-analysed the data set (combing the plot and trench sampling points) using the most commonly used temperature response functions (linear, exponential Q10 and modified Arrhenius function, see Figure 4 below).

5. The comparison of average soil respiration with other sites does not take temperature into account. It would be better to compare R10 or Q10 values. Alternatively you could use your know relation with temperature to adjust your values to the same temperature at other sites.

Response: We calculated the Q10 values and modified the results and discussion sections accordingly.

Results Section 3.3 "The linear temperature function explained around 44% of the temporal variation in total soil $CO_2$ efflux (Figure 4A, Table 3). Exponential (R2=0.13) and
modified Arrhenius ($R^2$=0.17) functions resulted in lower $R^2$ values (Table 3). The Q10 values for total soil $CO_2$ efflux was 1.6 (Table 3). A slightly stronger soil temperature response was found for heterotrophic respiration (linear function $R^2$=0.530, Figure 4B) with a Q10 value of 2.2 (Table 3). No significant relationship was found between soil temperature and autotrophic respiration (Figure 3C).

Neither a linear nor a quadratic function resulted in a significant relationship between SWC and total soil $CO_2$ efflux (Figure 4D). Heterotrophic respiration decreased significantly with increasing SWC ($R^2$=0.590, Figure 4E). In contrast a weak, but significant positive relationship was found between SWC and autotrophic respiration (Figure 4E). Bivariate polynomial functions did not result in higher $R^2$ values compared to univariate models (Table 3)."

Discussion Section 4.1: "While mean annual soil temperature partly explains the overall high mean soil $CO_2$ efflux measured in this forest, soil temperature was not a very good predictor of the temporal variation in total soil $CO_2$ efflux. Independent of the regression model used, soil temperature explained a small proportion (< 44%, Figure 4A, Table 3) of the seasonal variation in total soil $CO_2$ efflux. In temperate forest ecosystems in the Northern Hemisphere (Ngao et al., 2012; Bond-Lamberty and Tompson, 2014) soil temperature often explains more than 50% of the temporal variability in total soil $CO_2$ efflux. It is important to note that the soil temperature range in this kauri forest was narrow (around 7°C) compared to other temperate forests with a larger seasonal soil temperature amplitude (> 10°C, Paul et al., 2004). Thus, a seasonal temperature effect may not have been visible in this kauri forest. The Q10 value (1.6, $R^2$=0.172, Table 3) was at the lower end of the range reported for mixed and evergreen forests (Q10_10-20°C; 0.5-5.6; Bond-Lamberty and Tompson, 2014). However, low Q10 values have also been reported for other conifer forest, especially at sites characterized by mild winters (Borken et al. 2002; Curiel Yuste et al., 2004; Sulzman et al., 2005). Low Q10 values in evergreen forests have been explained by the lack of a distinct seasonality in photosynthesis and substrate supply (Curiel Yuste et al., 2004)."

6. The collars were inserted in November 2011 and efflux measurements commenced in January 2012. This does not leave enough time for the roots to decompose; therefore this is not truly a measurement of heterotrophic respiration. How was this accounted for and was there a decrease in RH over time as the roots decayed?

Response. We installed the collars in November 2011. Only measurements conducted after August 2012 have been included in this study. We did not correct our estimate of soil $CO_2$ efflux for decomposing root-derived $CO_2$ flux. We did not observe a significant insertion/trenching related change in heterotrophic respiration. We don't have data on kauri root decomposition but previous studies showed that kauri litter is characterized by very long residence times (between 9 and 78 years, Silvester and Orchard, 1999). To address the effect of root decomposition-derived $CO_2$ fluxes we included a statement in the methods section and modified the discussion as follows: "Cutting roots through inserting deep collars and trenching increases the dead root biomass (Heinemeyer et al., 2011). As we did not correct our estimates of soil $CO_2$ efflux for decomposing root-derived $CO_2$ fluxes the heterotrophic respiration may have been slightly overestimated (Hanson et al., 2000; Kuzyakov, 2006; Ngao et al., 2012)."

7. In the study site description, it would be good to know that the forest had been disturbed by tree removal and may not be in equilibrium.

Response: The following statement has been added in section 2.1 (Study site) "Four emergent kauri trees (up to 180 cm in DBH, approx.. 300 year-old) are found on the upper slope of the plot. At the lower slope tree fall and removal of five large kauri trees in the 1950s created gaps which are now dominated by a cohort of younger kauri trees".

L 187 states that efflux was measured on a number of days immediately after trenching, but this data are not presented.

Response: We removed this statement.

L 203 Where was this temperature measured, in the chamber?

Response: The temperature was measured next to (outside) the collar.

L 221 delete "of", also (45% C, 25 2.3% N) doesn't make sense.

Response: corrected (45% C, 2.3% N)

L 236 You state that there are two replicates, but on L 182 it says "one location"

Response: Thanks. We corrected the statement.

L 255 spelling of "Surfave"

Response: This has been changed.

L 263 Table 3 is referenced before Table 2

Response: Reference to Table 3 has been removed.

L 294 Please state which subplot is being referred to (Fig 2.?), also what is 14.2 +/- 0.1 a SD of a SEM.

Response: The subplot (Figure 2 A, B, C) has been included. Values are means $\pm$SE.

L 294 Please refer to the months as well as the season

Response: This has been changed

L 300 Fig 2A

Response: Changed accordingly

L 303 Change "locations" to times.

Response: This has been changed

L 313 I couldn't see an increase in variability during the dry summer of 2013 (I gather this is Jan – Mar 2013?)

Response: The statement has been modified. We included the monthly rainfall in Figure 2.

L 317 The data for summer/early autumn 2012 is not presented.

Response: This statement has been deleted.

L 329 Was SWC really affected by collar insertion, if so how?

Response: This section was deleted. Outside_Trench_Inserted data are no longer presented.

L 367 I am not sure how we can see in the table that there is a sign changes around 40% soil water content.

Response: We pooled all surface and inserted/trenched sampling points (plot and trench) and re-analysed the data (now shown in Figure 4D-E, see above).

L 377 Change Table 1 to Table 2

Response: Thanks, this has been corrected

L 381 Table 1 does not show the 0-30 cm values

Response: The root biomass values for 0-30 cm and total root biomass (organic layer plus mineral soil to 30 cm depth) have been added.

L 399 3.47 umols is 3.6 umol on line 303

Response: This has been changed.

L 458 Fig 2 the temperature difference is > 5 degrees" but Table 2 shows 6.6 degrees. Maybe there is a temperature response but it doesn't show up with such a small temperature range.

Response: This has been changed.

L 466 This sentence doesn't make sense.

Response: We re-phrased the statement.

L 482 Are the mature kauri at the site emergent? If so then state this in the site description.

Response: The mature kauri are emergent. The site description has been modified accordingly.

L 516 State early on that you are going to test the effect of collar depth on effluxes.

Response: See response comment 1.

Table 1 It would be good to have the litterfall summed over a year so it can be compared with other sites.

Response: Annual litterfall estimates are now included.

Table 2 I don't think it is necessary to show both the STD and the SE. Outside is misspelt as "Outsite_Trench_Surface". Using x, y and z for significant differences is confusing, stick with "c, d, e".

Response: We modified the table as suggested.

Table 3. Why are some numbers italicized? You need to define what a, b and c are.

Response: the italicized numbers indicated "significance". However, the table had been modified and p values are now included.

Fig 1 "Unknown"? – seems like it should be possible to get an identification of the tree species over the two years of the study, surely it can be classified as a broadleaf or not. The unknowns are filled circles, but open circles in the legend. There are two types of stars. How much are the size of the circles are scaled by the diameter? The 0.5 m contour lines are not really needed unless referred to in the manuscript. The plot has "trenched plots" as a title, this should be removed.

Response: We revised the figure.

Fig 2. These plots need to be labelled a, b, c. The sample points are joined up with lines; I cannot see why this can't be done in the SWC graph. The figure caption should indicate that these are means and state which soil efflux is being referred to (surface, trenched, inserted, plot or outside).

Response: The plots are now labelled A, B, C. the samples points for SWC cannot be joined up as measurements for two sampling dates are missing due to equipment failure. The figure caption has been revised as suggested.

Fig 4. Subplots should be labelled a-f. On subplot 4.2b the equation is wrong and ends up outside the 2nd x axis.

Response: The plots are now labelled A-F. The equations have been corrected.

L 984 is missing a ")", S is given in cm2 but the units along the x axis are in m2. Is m-2 m-2 correct? A is described as coefficient form, but does not appear in the equation.

Response: The equations have been corrected.

[Figure]

**Fig. 1.** Figure 4. Upper panel. Relationship between soil temperature and total soil CO2 efflux (A), heterotrophic respiration (B) and autotrophic respiration (C). Lower panel: Relationship between volumetric

---

## Author Response (AR1)

Dear editors,

we are pleased to submit a revised version of the manuscript soil-2016-21.

We thank the Topical Editor and reviewers for their comments and suggestions that helped to improve the quality of the manuscript and to clarify some aspects of the analysis.

We carefully revised the manuscript by addressing all the reviewers' comments. In particular we re-analysed the data to assess the soil temperature/soil water response of total $CO_2$ efflux, heterotrophic and autotrophic respiration.

Please find below the point-by-point response to the Topical Editor and reviewers' comments.

The authors' replies are in italics. Changes in the manuscript are highlighted in yellow.

Kind regards,
Luitgard Schwendenmann

**Topical Editor Decision: Revision (05 Jul 2016) by Dr. Axel Don**

I do not see that the insertion of the collars will significantly influence CO2 efflux 10 months after the insertion. Please add in l. 173 that the first 10 months after the insertion were not used to assess the CO2 efflux (as written in response to reviewer 3 comment 6).

*Response: We modified the statement (line 179-180)*

If models with different numbers of variables are going to compared (Q10 functions with 2 or 3 variables) the AIC should be used as indicator for the best model and not the RMSE or R2.

*Response: We included the AIC as an indicator for the best model (Table 3). Re-analysing the data we realized that the correlation coefficient (R2) of the linear temperature response function for total soil $CO_2$ efflux is 0.17 and not 0.44 as reported in our "author's response' letters. We corrected Table 3, Figure 4a and the text accordingly.*

Please provide the soil type in WRB classification for your study site (l. 147).

*Response: We included the soil type using the US soil classification (line 152-155, line 883-884).*

**Referee #1**

We thank the referee for the constructive comments which helped to improve the quality of the paper. Please find below a detailed response to the each of the comments.

General comments

The manuscript describes a study of soil respiration in a native forest of New Zealand. It is well written, meaning correct and fluent language, a clear introduction and presentation of the methods and results. It manages to describe well the characteristics of soil $CO_2$ efflux in this type of forest and has the advantage of being the first such study in this particular ecosystem.  The originality of the study is mostly if not entirely the result of this last point. While it presents correlation analyses and finds temperature and root biomass as the most important factors explaining the $CO_2$ fluxes, it remains otherwise mostly descriptive. Some results, interpretations and conclusions are not entirely convincing. In particular, I would question the correctness of the model fitting section.

*Response: We re-analysed the data set using the Q10 and modified Arrhenius (Lloyd and Taylor 1994) function to test the temperature response of total soil $CO_2$ efflux and heterotrophic (for details see response 2 regarding section 3.3)*

Specific comments

**1. Introduction** and Methods are well written. Here I find nothing to question. When trenching or inserting deep collars, severed roots can add to the decomposing pool and change the estimate of heterotrophic respiration. How were decomposing roots accounted for in this study?

*Response: We did not correct our estimate of soil $CO_2$ efflux for decomposing root-derived $CO_2$ flux. We did not observe a significant insertion/trenching related change in heterotrophic respiration. We don't have data on kauri root decomposition but previous studies showed that kauri litter is characterized by very long residence times (between 9 and 78 years, Silvester and Orchard, 1999). To address the effect of root decomposition-derived $CO_2$ fluxes we included a statement in the methods section (line 262-264) and modified the discussion as follows (line 537-541): "Cutting roots through inserting deep collars and trenching increases the dead root biomass (Heinemeyer et al., 2011). As we did not correct our estimates of soil $CO_2$ efflux for decomposing root-derived $CO_2$ fluxes the heterotrophic respiration may have been slightly overestimated (Hanson et al., 2000; Kuzyakov, 2006; Ngao et al., 2012)."*

**2. In section 3.3** you describe fitting models for the T response but fail to mention the most common used i.e. Q10 or LT, etc. The Q10-function is usually equivalent to an exponential function and has only 2 parameters, i.e. a * Q10^((T-Tref)/10). Why do you have 3 parameters for the Q10 function? Is one a constant? Please check your functions in Table 3. Everything in the exponent should be closed by parenthesis. Also, the fact that you improved your $R^2$ with a bivariate model but have much larger RMSE is not

consistent. Check that your calculations are correct. Adding explanatory variables should only reduce the RMSE if you are using the same data.

It would be good to have plots showing the response to T and M.

*Response: The Q10 function (Schlentner and van Cleve, 1985) we used is a sigmoid function with three parameters (a = lower limit of soil $CO_2$ efflux, a+1/b = maximum flux, c = Q10 related parameter).*

*Based on the recommendation by referee 1 and referee 3 we re-analysed our data set using the most commonly used temperature response functions (linear, exponential Q10 and modified Arrhenius function). The linear temperature response function provided the best fit, explaining 17% of total soil $CO_2$ efflux and 47% of the heterotrophic respiration (see Figure below). This new figure (Figure 4) will be included in the manuscript. The methods, results and discussion sections have been modified as follows:*

*Methods Section 2.5 Data analysis (line 269-279): "Univariate and bivariate models were used to investigate the relationship between total soil $CO_2$ efflux, heterotrophic and autotrophic respiration and the abiotic factors soil temperature and volumetric soil water content. Data from within the research plot and trench sampling points were combined. The temperature response of soil $CO_2$ efflux was tested using a linear, exponential ($Q_{10}$, van't Hoff 1898) and modified Arrhenius function (Lloyd and Taylor, 1994). Linear and quadratic functions were used to assess the soil water dependence of soil $CO_2$ efflux. The combined effect of soil temperature and soil water content on soil $CO_2$ efflux was tested using a polynomial function. Coefficient of determination ($R^2$), standard error of estimate (SEE), and Akaike Information Criterion (AIC) were used to evaluate model performance. The analysis was conducted using Sigma Plot Sigma Plot (Version 13, Systat Software Inc., Chicago, IL, USA).*

*Results Section 3.3 (line 358-370): "Independent of the model used, soil temperature explained less than 20% of the temporal variation in total soil $CO_2$ efflux (Fig. 4a, Table 3). The $Q_{10}$ values for total soil $CO_2$ efflux was 1.6 (Table 3). A slightly stronger soil temperature response was found for heterotrophic respiration (Fig. 4b, Table 3) with a $Q_{10}$ value of 2.2 (Table 3). However, all temperature models for heterotrophic respiration had higher AIC values compared to total soil $CO_2$ efflux (Table 3) which suggests a poorer performance. No significant relationship was found between soil temperature and autotrophic respiration (Fig. 4c, Table 3).*

*Neither a linear nor a quadratic function resulted in a significant relationship between SWC and total soil $CO_2$ efflux (Fig. 4d, Table 3). Heterotrophic respiration decreased significantly with increasing SWC (Fig. 4e, Table 3). In contrast a weak, but significant quadratic relationship was found between SWC and autotrophic respiration (Fig. 4f).*

*Bivariate polynomial functions did not result in higher $R^2$ or better AIC values compared to univariate models (Table 3)."*

*Discussion Section 4.1 (line 451-466): "While mean annual soil temperature partly explains the overall high mean soil $CO_2$ efflux measured in this forest, soil temperature was not a very good predictor of the temporal variation in total soil $CO_2$ efflux.*

*Independent of the regression model used, soil temperature explained a small proportion (< 20%, Fig. 4a, Table 3) of the seasonal variation in total soil $CO_2$ efflux. In temperate forest ecosystems in the Northern Hemisphere (Ngao et al., 2012; Bond-Lamberty and Tompson, 2014) soil temperature often explains more than 50% of the temporal variability in total soil $CO_2$ efflux. It is important to note that the soil temperature range in this kauri forest was narrow (around 7°C) compared to other temperate forests with a larger seasonal soil temperature amplitude (> 10°C, Paul et al., 2004). Thus, a seasonal temperature effect may not have been visible in this kauri forest. The $Q_{10}$ value (1.6, Table 3) was at the lower end of the range reported for mixed and evergreen forests ($Q_{10\_10-20°C}$; 0.5-5.6; Bond-Lamberty and Tompson, 2014). However, low $Q_{10}$ values have also been reported for other conifer forest, especially at sites characterized by mild winters (Borken et al. 2002; Curiel Yuste et al., 2005b; Sulzman et al., 2005). Low $Q_{10}$ values in evergreen forests have been explained by the lack of a distinct seasonality in photosynthesis and substrate supply (Curiel Yuste et al., 2005b).*

*"*

[Figure]

*Upper panels. Relationship between soil temperature and total soil $CO_2$ efflux (a), heterotrophic respiration (b) and autotrophic respiration (c). Lower panels: Relationship between soil volumetric water content and total soil $CO_2$ efflux (d), heterotrophic respiration (e) and autotrophic respiration (f). Regression lines are only displayed for significant linear relationships. The results for other uni- and bivariate functions are shown in Table 3.*

**3. In the discussion** you calculate an average and compare with other ecosystems. Using the average of your measurements is incorrect. Since there is a T and root effect you should account for these when getting yearly estimates. At least use the T

relationship, since T at night is probably lower, so the yearly average is lower than that of your measurements.

*Response: The yearly estimate was re-calculated using the linear response function (best fit) and soil temperature data (5 cm depth, 30 min averages). The revised annual estimate resulted in a slightly higher annual estimate (1324 ± 121 g C m$^{-2}$ yr$^{-1}$) (line 252-254, line 396-397).*

**4. You discuss how the vegetation** may control the amount of $CO_2$ efflux. The question of whether your system is near equilibrium is important here. If a forest is near equilibrium, the quality of the litter is important only in determining the stock sizes, not the $CO_2$ fluxes. The latter will be equal to the amounts of input.

*Response: The forest stand is dominated by a few emergent (up to 300 year-old) kauri trees. It is unlikely that this forest in near equilibrium as tree fall and removal of five large kauri trees in the 1950s created gaps which are now dominated by a cohort of younger kauri trees. The following statement has been added (line 141-144).*

*2.1 Study site "Kauri tree size distribution differs within the plot. Four emergent kauri trees (up to 180 cm in DBH, approx. 300 year-old) are found on the upper slope of the plot. At the lower slope tree fall and removal of five large kauri trees in the 1950s created gaps which are now dominated by a cohort of younger kauri trees."*

**5. When discussing the effect of T**, make clear that your T range is small, which does not mean there is little T effect, just that you cannot see it. In terms of the average yearly T, this will probably have a larger effect in how it affects the productivity of the vegetation, so indirectly through litter input.

*Response: The discussion on the effect of temperature on soil $CO_2$ efflux has been modified (see above, Response 2, line 451-466)*

**6. In conclusions, you state** that the study has found that the vegetation type exerts a strong influence on soil carbon related processes. This is an effect of all land vegetation and is no finding by itself, thus making for a very weak conclusion. An insight on the vegetation effect on the soil C stocks or some other more specific observation should come here. Also, you mention that species effects were should in the study, however no species comparison was made, so mentioning species effects is incorrect, here and in the abstract.

*Response: We found strong relationships between the "index of local contribution" (a measure of kauri tree size and distribution) and total soil $CO_2$ efflux, root biomass and mineral C:N ratio suggesting that the spatial arrangement of kauri trees influences soil characteristics and soil $CO_2$ efflux. This is in line with previous findings showing that kauri trees exert a strong control on soil pH and soil nitrogen (Silvester 2000; Jongkind et al. 2007; Verkaik et al. 2007; Wyse et al., 2014). In our conclusion we wanted to highlight the importance of investigating biotic factors (here: kauri tree distribution and size as a*

*measure of forest structure) in soil carbon related studies. The abstract and conclusion sections have been revised accordingly (line 29-30, line 599-600)*

**7. Lines 317-319** This line is not clear to me

*Response: This statement has been deleted.*

**Referee #2**

We thank the referee for providing helpful comments. Please find below a detailed response to the each of the comments

General remarks:

The interesting manuscript (ms) represents the investigations within a native forest in New Zealand with the aim to characterize dependencies between forest structure, soil properties, meteorological conditions and soil respiration processes. The ms has clear objectives and represents a good contribution to scientific progress in the interdisciplinary field of abiotic and biotic soil respiration influences. In order to describe temporal variability the data interpretation is based on times series over 18 month of $CO_2$ efflux measurements. Furthermore, the authors considered a huge number of relevant references, giving a comprehensive insight and the chance to compare the approach with the results from other researchers. The overall quality of the manuscript is high. It offers interesting insights into the soil $CO_2$ efflux within an old-growth kauri forest and the main controlling factors for such a forest site. The entire results are discussed based on sound statistical analysis. However, in my opinion there are some issues which need to be discussed more in detail to close some minor gaps improving the ms.

Specific comments:

**Line 36:** Janssens et al., 2001 – two times in reference list – which one is meant here? → 2001a, b

*Response: Janssens et al, 2001a; in-text citations (line 36) and reference list (line 760) have been changed accordingly*

**Line 138 and Figure 1:** The figure 1 is not easy to understand concerning the experimental set up. There are some items, which are not explained in the legend: filled grey circles, filled grey stars, grey lines (I supposed it is the topography in m a.s.l.?) Where are the trenched plots? There is only text at the upper edge.

*Response: The figure and legend have been modified to provide a better illustration of the experimental set-up.*

**In the context topography**: you mention a potential dependency between topography and organic layer thickness (line 448-450). I would strongly recommend analyzing a functional trend between soil moisture and topography and hence, maybe some influences on soil respiration. From your data compilation it is not clear to see, but maybe the soil moisture differences you mentioned in Table 2 for the trenched plots could be superimposed by topographic driven soil moisture differences. You

can find some discussions in recent papers, e.g.,

Masamichi Takahashi, Keizo Hirai, Pitayakon Limtong, Chaveevan Leaungvutivirog, Samreong Panuthai, Songtam Suksawang, Somchai Anusontpornperm & Dokrak Marod (2011) Topographic variation in heterotrophic and autotrophic soil respiration in a tropical seasonal forest in Thailand, Soil Science and Plant Nutrition, 57:3, 452-465, DOI: 10.1080/00380768.2011.589363

Wang, B., Zha, T.S., Jia, X., Gong, J.N., Wu, B., Bourque, C.P.A., Zhang, Y., Qin, S.G., Chen, G.P., Peltola, H., 2015. Microtopographic variation in soil respiration and its controlling factors vary with plant phenophases in a desert–shrub ecosystem. Biogeosciences 12, 5705-5714.

*Response: Following the referee's recommendation we investigated the relationship between elevation (m a.s.l.), soil moisture, organic layer thickness, root biomass and total soil $CO_2$ efflux. We found a significant negative correlation between elevation and organic layer thickness (r=-0.539, p=0.021). However, none of the other parameters showed a significant relationship with elevation. We modified the discussion (Section 4.1) as follows (line 440-450): "The topography of the study site (moderately to steep slope) likely explains the negative correlation between organic layer thickness and elevation (r=-0.539, p=0.021). Erosive removal of the organic layer and mineral soil on steep slopes and deposition downslope have been shown to affect soil characteristics and C cycling (Quideau, 2002; Vitousek et al., 2003; Yoo et al., 2005). For example, in a temperate forest in Japan (Nakane et al., 1994) and a tropical seasonal forest in Thailand (Takahashi et al., 2011) soil $CO_2$ efflux decreased with increasing slope. However, we did not find any correlation between elevation and total soil $CO_2$ efflux, root biomass, and soil moisture suggesting that forest structure (see 4.2) may have had a stronger effect on soil characteristics than topography at this site."*

**Line 221**: . . . plant material (45% C, 25 2.3% N) . . . → 25 ??

*Response: corrected (45% C, 2.3% N) (line 227)*

**Line 312 and Figure 2:** You mentioned a relation between high $CO_2$ efflux and heavy rain events (as described and shown in paper Macinnis-Ng & Schwendenmann, 2015). Why do not show precipitation information in figure 2? I suppose, that graph would visually support very well your interpretation!

*Response: Figure 2 was modified as suggested.*

**Line 333:** . . .Outside_Trench_Insered . . . t is missing

*Response: this section was deleted*

**Line 537:** reference Epron et al., 2001 is not in reference list

*Response: added Epron et al., 2001 (line 705-707)*

**Line 564:** *. . .*de Jong and Schappert*. . .*

*Response: corrected – Schappert (line 575)*

**Line 957:** January

*Response: spelling mistake corrected (line 982)*

**Line 958:** The different letters .. indicates a significant difference *. . .* between what? Mean and Median? What means a, b, x, y, z?

*Response: The table caption has been changed as follows: "Samples were separated into plot and trench for the statistical analysis due to different sampling designs. Different letters (a, b for plot; c, d for trench) for a given variable indicate a significant difference between treatments." (line 980-984)*

**Line 965a:** The determined regression coefficients are in all cases very weak – hence, it is not really a convincing correlation! As an example, you could include a figure to show the different modelled approaches.

*Response: We re-analysed the data set (combing the plot and trench sampling points) using the most commonly used temperature response functions (linear, exponential Q10 and modified Arrhenius function). This results of the linear regression (best fit) are shown in Figure 4 which will be included in the manuscript.*

[Figure]

*Upper panels. Relationship between soil temperature and total soil $CO_2$ efflux (a), heterotrophic respiration (b) and autotrophic respiration (c). Lower panels: Relationship between soil volumetric water content and total soil $CO_2$ efflux (d), heterotrophic respiration (e) and autotrophic respiration (f). Regression lines are only displayed for significant linear relationships. The results for other uni- and bivariate functions are shown in Table 3.*

**Line 965b:** Table 3 subscription ... adjusted R2 = coefficient... RMSE = root mean aquare

*Response: changed (Table 3)*

**Line 798**: reference Metcalfe et al., 2011 is not mentioned in text

*Response: Metcalfe et al., 2011 added in Section 1 (line 62)*

**Referee #3**

We highly appreciate the referee's comments and suggestions, which helped improve the quality of the manuscript. Please find below a detailed response to the each of the comments.

General comments

This study reports measurements of soil respiration and their component parts in a kauri forest in New Zealand. Both trenching and statistical techniques are used to partition the total soil efflux into auto and heterotrophic respiration. Statistical methods are then used to investigate the temporal controls by environmental parameters. Tree root biomass and "tree influence" are used to look for controls on the spatial variability in soil $CO_2$ efflux. There are extensive references and good comparisons with data from the NH. The interest in this manuscript doesn't so much lie in its respiration results and partitioning, but rather its combination of soil respiration and spatial patterns relating to root biomass.

Specific comments

**1.** An attempt was made to test soil respiration methodology with comparisons between surface and inserted rings. However, this was not well described in the introduction and I was confused to why they did this.

*Response: We revised section 1 as follows (line 121-130): "The specific objectives of our study were: (i) to quantify total soil $CO_2$ efflux, (ii) to test the effect of collar insertion depth on soil $CO_2$ efflux, (iii) partition total soil $CO_2$ efflux into autotrophic and heterotrophic respiration, (iv) to identify the factors controlling the temporal variation of total soil $CO_2$ efflux and its component fluxes, and (v) to test the effect of kauri tree size and distribution on total soil $CO_2$ efflux and soil properties. We used direct (trenching) and indirect (regression technique) approaches to partition total soil $CO_2$ efflux into the autotrophic and heterotrophic components. Given that old-growth kauri forests are often characterised by thick organic layers, deep collars were deployed to assess the effect of insertion depth on total soil $CO_2$ efflux and to quantify to proportion of autotrophic and heterotrophic respiration in this layer"*

**2.** In the Experimental Setup the reason for the inserted and surface chambers should be explained.

*Response: We revised the methods section to improve the links between objectives and the experimental set up as follows (line 121-130). We decided not to include the data from the "Outside_Trench_Inserted" sampling points as the results confirmed the findings in the plot (17% reduction in total $CO_2$ efflux due to deep collar insertion).*

*2.2. Experimental set-up (line 170-181) "To measure the effect of collar insertion depth and to quantify the proportion of autotrophic and heterotrophic respiration to total soil $CO_2$ efflux in the organic layer, a cluster of three 'deep' PVC collars (10 cm in diameter,*

*20 cm in height) was inserted next to each sampling point for surface soil $CO_2$ efflux measurements. Three collars per cluster were spaced evenly around the circumference of a circle 2 m in diameter, with small adjustments in the spacing to avoid large roots. Each collar was driven right through the organic layer and 1-2 cm into the mineral soil layer to cut off the roots growing in the organic layer. In order to prevent $CO_2$ uptake, any vegetation inside the collars was regularly removed. The thickness of the organic layer at each grid point was measured using a ruler outside each collar. The deep collars were inserted in November 2011 and left in place over the measurement period. Efflux was measured from August 2012 (9 months after insertion) to January 2014. Here after, these sampling points are known as Plot_Inserted."*

*Line 182-195: "We used the trenching approach to separate heterotroph and autotrophic respiration in the organic layer plus mineral soil to 30 cm depth. To avoid disturbing the long-term research plot the trenching experiment was set-up directly adjacent (upslope) to the research plot. In July 2012, six 2 x 2 m plots were trenched to 30 cm depth based on a preliminary study showing that the majority of fine roots (over 80%) are located in the organic layer and top 30 cm of the mineral soil. The trenches were double-lined with a water permeable polypropylene fabric and backfilled. During trenching, trampling and disturbance inside the 2 x 2 m plots were avoided as far as possible. Two types of measurements were conducted. First, total soil $CO_2$ efflux was measured at two sampling points outside each trenched plot (Outside_Trench_Surface, n=12) in the same way as the Plot_Surface samples were measured (see above). Second, two collars were randomly placed inside the trenched plot (Trench_Inserted). The collors were inserted 1-2 cm into the mineral soil layer (deep collars) as described above. Soil $CO_2$ efflux was measured bi-weekly to monthly from August 2012 until December 2013."*

**3.** I found the overall aims of the manuscript confusing and this wasn't helped by the description of the methods and the 5 different types of soil surface measurements. A better way to arrange this might be to describe each aim and then the methods that go with it. Eg Collar insertion depth, respiration partition, annual soil respiration, spatial variability vs temporal variability.

*Response: We modified the methods section as suggested (see response 2). We decided not to include the data from the "Outside_Trench_Inserted" sampling points as the results confirmed the findings in the plot (17% reduction in total $CO_2$ efflux due to deep collar insertion).*

**4.** It would be good to see the data, especially the relationship between soil respiration (total, Ra, Rh) and temperature.

*Response: A figure showing the relationship between total soil $CO_2$ efflux, heterotrophic and autotrophic respiration has been included. We re-analysed the data set (combing the plot and trench sampling points) using the most commonly used temperature response functions (linear, exponential Q10 and modified Arrhenius function).*

[Figure]

*Upper panels. Relationship between soil temperature and total soil CO₂ efflux (a), heterotrophic respiration (b) and autotrophic respiration (c). Lower panels: Relationship between soil volumetric water content and total soil CO₂ efflux (d), heterotrophic respiration (e) and autotrophic respiration (f). Regression lines are only displayed for significant linear relationships. The results for other uni- and bivariate functions are shown in Table 3.*

**5.** The comparison of average soil respiration with other sites does not take temperature into account. It would be better to compare R10 or Q10 values. Alternatively you could use your know relation with temperature to adjust your values to the same temperature at other sites.

*Response: We calculated the Q₁₀ values and modified the results and discussion sections accordingly.*

*Results Section 3.3 (line 358-370): "Independent of the model used, soil temperature explained less than 20% of the temporal variation in total soil CO₂ efflux (Fig. 4a, Table 3). The Q₁₀ values for total soil CO₂ efflux was 1.6 (Table 3). A slightly stronger soil temperature response was found for heterotrophic respiration (Fig. 4b, Table 3) with a Q₁₀ value of 2.2 (Table 3). However, all temperature models for heterotrophic respiration had higher AIC values compared to total soil CO₂ efflux (Table 3) which suggests a poorer performance. No significant relationship was found between soil temperature and autotrophic respiration (Fig. 4c, Table 3).*

*Neither a linear nor a quadratic function resulted in a significant relationship between SWC and total soil CO₂ efflux (Fig. 4d, Table 3). Heterotrophic respiration decreased significantly with increasing SWC (Fig. 4e, Table 3). In contrast a weak, but significant*

*quadratic relationship was found between SWC and autotrophic respiration (Fig. 4f).*

*Bivariate polynomial functions did not result in higher $R^2$ or better AIC values compared to univariate models (Table 3)."*

*Discussion Section 4.1 (line 451-466): "While mean annual soil temperature partly explains the overall high mean soil $CO_2$ efflux measured in this forest, soil temperature was not a very good predictor of the temporal variation in total soil $CO_2$ efflux. Independent of the regression model used, soil temperature explained a small proportion (< 20%, Fig. 4a, Table 3) of the seasonal variation in total soil $CO_2$ efflux. In temperate forest ecosystems in the Northern Hemisphere (Ngao et al., 2012; Bond-Lamberty and Tompson, 2014) soil temperature often explains more than 50% of the temporal variability in total soil $CO_2$ efflux. It is important to note that the soil temperature range in this kauri forest was narrow (around 7°C) compared to other temperate forests with a larger seasonal soil temperature amplitude (> 10°C, Paul et al., 2004). Thus, a seasonal temperature effect may not have been visible in this kauri forest. The $Q_{10}$ value (1.6, Table 3) was at the lower end of the range reported for mixed and evergreen forests ($Q_{10\_10-20°C}$; 0.5-5.6; Bond-Lamberty and Tompson, 2014). However, low $Q_{10}$ values have also been reported for other conifer forest, especially at sites characterized by mild winters (Borken et al. 2002; Curiel Yuste et al., 2005b; Sulzman et al., 2005). Low $Q_{10}$ values in evergreen forests have been explained by the lack of a distinct seasonality in photosynthesis and substrate supply (Curiel Yuste et al., 2005b).*

**6.** The collars were inserted in November 2011 and efflux measurements commenced in January 2012. This does not leave enough time for the roots to decompose; therefore this is not truly a measurement of heterotrophic respiration. How was this accounted for *and was there a decrease in RH over time as the roots decayed?*

*Response. We installed the collars in November 2011. Only measurements conducted after August 2012 have been included in this study. We did not correct our estimate of soil $CO_2$ efflux for decomposing root-derived $CO_2$ flux. We did not observe a significant insertion/trenching related change in heterotrophic respiration. We don't have data on kauri root decomposition but previous studies showed that kauri litter is characterized by very long residence times (between 9 and 78 years, Silvester and Orchard, 1999). To address the effect of root decomposition-derived $CO_2$ fluxes we included a statement in the methods section (line 262-264) and modified the discussion as follows (line 537-541): "Cutting roots through inserting deep collars and trenching increases the dead root biomass (Heinemeyer et al., 2011). As we did not correct our estimates of soil $CO_2$ efflux for decomposing root-derived $CO_2$ fluxes the heterotrophic respiration may have been slightly overestimated (Hanson et al., 2000; Kuzyakov, 2006; Ngao et al., 2012)."*

**7.** In the study site description, it would be good to know that the forest had been disturbed by tree removal and may not be in equilibrium.

*Response: The following statement has been added in section 2.1 (Study site, line 141-144) "Four emergent kauri trees (up to 180 cm in DBH, approx. 300 year-old) are found*

*on the upper slope of the plot. At the lower slope tree fall and removal of five large kauri trees in the 1950s created gaps which are now dominated by a cohort of younger kauri trees".*

**L 187 states** that efflux was measured on a number of days immediately after trenching, but this data are not presented.

*Response: We removed this statement.*

**L 203** Where was this temperature measured, in the chamber?

*Response: The temperature was measured next to (outside) the collar (line 214)*

**L 221** delete "of", also (45% C, 25 2.3% N) doesn't make sense.

*Response: corrected (45% C, 2.3% N) (line 227)*

**L 236** You state that there are two replicates, but on L 182 it says "one location"

*Response: Thanks. We corrected the statement.*

**L 255** spelling of "Surfave"

*Response: This has been changed.*

**L 263** Table 3 is referenced before Table 2

*Response: Reference to Table 3 has been removed.*

**L 294** Please state which subplot is being referred to (Fig 2.?), also what is 14.2 +/- 0.1 a SD of a SEM.

*Response: Reference to subplots (Figure 2 a, b, c) has been included. Values are means ±SE.*

**L 294** Please refer to the months as well as the season

*Response: This has been changed (line 307-310)*

**L 300** Fig 2A

*Response: Changed accordingly*

**L 303** Change "locations" to times.

*Response: This has been changed*

**L 313** I couldn't see an increase in variability during the dry summer of 2013 (I gather this is Jan – Mar 2013?)

*Response: The statement has been modified. We included fortnightly rainfall in Figure 2c.*

**L 317** The data for summer/early autumn 2012 is not presented.

*Response: This statement has been deleted.*

**L 329** Was SWC really affected by collar insertion, if so how?

*Response: This section was deleted. Outside_Trench_Inserted data are no longer presented.*

**L 367** I am not sure how we can see in the table that there is a sign changes around 40% soil water content.

*Response: We pooled all surface and inserted/trenched sampling points (plot and trench) and re-analysed the data (now shown in Figure 4d-f).*

**L 377** Change Table 1 to Table 2

*Response: Thanks, this has been corrected*

**L 381** Table 1 does not show the 0-30 cm values

*Response: The root biomass values for 0-30 cm and total root biomass (organic layer plus mineral soil to 30 cm depth) have been added.*

**L 399** 3.47 umols is 3.6 umol on line 303

*Response: This has been changed.*

**L 458** Fig 2 the temperature difference is > 5 degrees" but Table 2 shows 6.6 degrees. Maybe there is a temperature response but it doesn't show up with such a small temperature range.

*Response: This has been changed (line 458)*

**L 466** This sentence doesn't make sense.

*Response: We re-phrased the statement.*

**L 482** Are the mature kauri at the site emergent? If so then state this in the site description.

*Response: The mature kauri are emergent. The site description has been modified* accordingly. (line 141-144)

**L 516** State early on that you are going to test the effect of collar depth on effluxes.

*Response: See response comment 1.*

**Table 1** It would be good to have the litterfall summed over a year so it can be compared with other sites.

*Response: Annual litterfall estimates are now included.(Table 1)*

**Table 2** I don't think it is necessary to show both the STD and the SE. Outside is misspelt as "Outsite_Trench_Surface". Using x, y and z for significant differences is confusing, stick with "c, d, e".

*Response: We modified the table as suggested.*

**Table 3.** Why are some numbers italicized? You need to define what a, b and c are.

*Response: the italicized numbers indicated "significance". However, the table had been modified and p values are now included.*

**Fig 1 "Unknown"?** – seems like it should be possible to get an identification of the tree species over the two years of the study, surely it can be classified as a broadleaf or not. The unknowns are filled circles, but open circles in the legend. There are two types of stars. How much are the size of the circles are scaled by the diameter? The 0.5 m contour lines are not really needed unless referred to in the manuscript. The plot has "trenched plots" as a title, this should be removed.

*Response: We revised the figure.*

**Fig 2.** These plots need to be labelled a, b, c. The sample points are joined up with lines; I cannot see why this can't be done in the SWC graph. The figure caption should indicate that these are means and state which soil efflux is being referred to (surface, trenched, inserted, plot or outside).

*Response: The plots are now labelled a, b, c. the samples points for SWC cannot be*

*joined up as measurements for two sampling dates are missing due to equipment failure. The figure caption has been revised as suggested.*

**Fig 4.** Subplots should be labelled a-f. On subplot 4.2b the equation is wrong and ends up outside the 2nd x axis.

*Response: The plots are now labelled a-f. The equations have been corrected.*

**L 984** is missing a ")", S is given in cm2 but the units along the x axis are in m2. Is m-2 m-2 correct? A is described as coefficient form, but does not appear in the equation.

*Response: The equations have been corrected.*